# IMAGE-FREE ZERO-SHOT LEARNING VIA ADAPTIVE SEMANTIC-GUIDED CLASSIFIER INJECTION

## ABSTRACT

*Zero-Shot Learning* (ZSL) aims to classify images from *unseen* classes by leveraging semantic relationships with *seen* classes. Most ZSL methods require access to visual data for training or adaptation, limiting their applicability in image-free scenarios. *Image-free Zero-Shot Learning* (I-ZSL) addresses this challenge by enabling pre-trained models to recognize unseen classes without image data. However, existing I-ZSL approaches rely on pre-defined class descriptions and task-agnostic text encoders, which often fail to capture domain-specific semantics. We propose *Adaptive Semantic-Guided Classifier Injection* (ASCI), a novel I-ZSL framework that eliminates reliance on manually curated descriptions. ASCI leverages large language models to generate class-pair affinity descriptions, capturing structured relationships between seen and unseen classes. A trainable text encoder refines these descriptions, ensuring alignment with task-specific semantics. Dynamically computed affinity scores guide the injection of robust classifiers for unseen classes while preserving the structural consistency of the pre-trained classification space. Experiments on benchmark datasets demonstrate that ASCI outperforms existing I-ZSL methods, particularly in fine-grained classification tasks.

## 1 INTRODUCTION

AI models have achieved remarkable success in extracting rich visual and semantic knowledge from large-scale datasets (LeCun et al., 2010; 2015). However, the increasing restrictions on data sharing due to privacy, copyright, and security concerns pose significant challenges (Papernot et al., 2018). These issues are particularly critical in sensitive domains such as healthcare and security, where access to data is tightly regulated. Additionally, proprietary and industrial datasets are often inaccessible to researchers, limiting the ability to train and deploy models across different organizations. As a result, these constraints hinder the transfer of knowledge to novel tasks, reducing the broader applicability of AI models (Xian et al., 2019; Wang et al., 2019; Zhong et al., 2025).

To address these challenges, *Zero-Shot Learning* (ZSL) (Mensink et al., 2014; Romera-Paredes & Torr, 2015) has emerged as a promising alternative. ZSL enables models to recognize unseen classes by leveraging semantic relationships between seen and unseen categories, rather than requiring labeled training samples. This capability is especially valuable in real-world applications where collecting labeled data for every possible category is infeasible (Pourpanah et al., 2023). However, most ZSL approaches rely on access to visual data or models during training or adaptation (Tang et al., 2024; Chen et al., 2024), limiting their usability in image-free scenarios where neither seen nor unseen class images are available. This restriction is particularly problematic in domains with strict privacy regulations or environments where data collection is expensive and time-consuming.

To overcome these limitations, Christensen et al. (2023) introduce the task of *Image-free Zero-Shot Learning* (I-ZSL). Given a pre-trained model and a classification task, I-ZSL extends the model to include unseen categories without requiring access to any images or models. The proposed ICIS model addresses this problem using two key components: (*i*) *pre-defined class descriptions*, which provide semantic representations for unseen classes, and (*ii*) *pre-trained text encoders*, which map these descriptions into a shared semantic space with classifier weights (Christensen et al., 2023). While effective, this approach presents two fundamental limitations: (*1*) Pre-defined descriptions assume the availability of accurate and detailed semantic annotations, which is often infeasible, particularly in domains requiring fine-grained category distinctions (Wah et al., 2011). (*2*) Pre-trained

text encoders, typically trained on general-purpose corpora, fail to capture task-specific semantics, leading to suboptimal alignment between visual and semantic spaces. Moreover, these text encoders do not adapt to the classification task, reducing their effectiveness in specialized domains where class relationships are complex and context-dependent. These challenges highlight the need for a more adaptive and task-specific I-ZSL framework capable of functioning effectively in applications.

In this work, we propose *Adaptive Semantic-Guided Classifier Injection* (ASCI), a novel framework designed for I-ZSL. Unlike previous work, ASCI eliminates the dependency on pre-defined descriptions and instead leverages *large language models* to generate *class-pair affinity descriptions* that capture structured relationships between seen and unseen classes. These descriptions provide a richer semantic context, enabling better generalization to unseen categories. Furthermore, ASCI introduces a trainable text encoder that adaptively aligns affinity descriptions with task-specific semantics, addressing the shortcomings of static, pre-trained encoders. By dynamically refining class-pair affinity scores during training, ASCI ensures that injected classifiers for unseen classes remain robust, well-aligned, and effective, even in challenging classification scenarios.

We evaluate ASCI on benchmarks under conventional and generalized ZSL settings. Experimental results demonstrate that ASCI consistently outperforms existing I-ZSL methods, particularly in fine-grained classification tasks where subtle inter-class differences must be captured. Ablation studies further validate the contributions of adaptive encoding and class-pair affinity descriptions in improving model scalability and generalization.

## 2 RELATED WORK

**Zero-Shot Learning.** ZSL has emerged as a promising paradigm for handling classification tasks where labeled data for certain categories is unavailable (Norouzi et al., 2014; Mensink et al., 2014; Romera-Paredes & Torr, 2015). By leveraging semantic information and image visual features, ZSL models establish relationships between seen and unseen classes, enabling the transfer of knowledge to novel tasks (Xian et al., 2017; Schönfeld et al., 2019; Pourpanah et al., 2023; Chen et al., 2023). Common approaches utilize either visual attribute-based representations (Farhadi et al., 2009; Lampert et al., 2014; Hou et al., 2024) or manually designed class description embeddings (Socher et al., 2013; Xian et al., 2019; Wang et al., 2023; Qu et al., 2025) to define semantic spaces. However, these methods often rely on high-quality annotated data for seen classes or assume the availability of auxiliary datasets, which are not always accessible in sensitive or resource-constrained domains.

**Image-Free Zero-Shot Learning.** In addition, conventional ZSL methods heavily depend on *visual data or models* during training or adaptation (Tang et al., 2024; Chen et al., 2024), which limits their feasibility in scenarios where data sharing is restricted due to privacy or security concerns (Papernot et al., 2018). To address these limitations, Christensen et al. (2023) introduce *Image-free Zero-Shot Learning* (I-ZSL), which eliminates the need for images from both seen and unseen classes. The ICIS model (Christensen et al., 2023) employs predefined class descriptions and pre-trained text encoders to map semantic representations of unseen classes into a shared space with classifier weights. Despite its effectiveness, ICIS faces three main limitations: (*1*) it relies on predefined class descriptions, which assumes access to detailed annotations for unseen classes—an unrealistic requirement in fine-grained or specialized domains; (*2*) it models semantics only at the class level, ignoring pairwise relationships that are often crucial for distinguishing between semantically similar classes; and (*3*) it uses a fixed, pre-trained text encoder, which limits the model's ability to adapt semantic representations to the task. In addition, ICIS provides no theoretical analysis on the quality of synthesized classifiers or the effects of semantic alignment and distributional variance.

**Large Language Models for Semantic Representations.** Recent advancements in *Large Language Models* (LLMs) have shown their potential in generating rich, context-aware semantic representations (Brown et al., 2020; Radford et al., 2021). These models have been applied to various natural language processing and cross-modal learning tasks, where they bridge gaps between text and other modalities (Xu et al., 2022; Tang et al., 2024). LLMs typically operate on general-purpose corpora and are not optimized for generating task-specific representations required for fine-grained classification tasks (Christensen et al., 2023). Esfandiarpoor & Bach (2024) propose using LLMs to generate descriptions to understand the correlation between visual data and target classes. However, visual data still manipulates this process, and the possibilities of I-ZSL still remain underexplored.

Table 1: Model comparison of ASCI and related work.

| Model | I-ZSL | w/o Manual Defined Class Description | Adaptive Semantic Guidance | w/o Image Vision Features |
|---|---|---|---|---|
| ConSE (Norouzi et al., 2014) | ✔ | ✘ | ✘ | ✔ |
| COSTA (Mensink et al., 2014) | ✔ | ✘ | ✘ | ✔ |
| Sub.Reg. (Akyürek et al., 2022) | ✘ | ✔ | ✘ | ✘ |
| wDAE (Gidaris & Komodakis, 2019) | ✘ | ✔ | ✘ | ✘ |
| WAvg (Xu et al., 2022) | ✘ | ✘ | ✔ | ✘ |
| SMO (Xu et al., 2022) | ✘ | ✘ | ✔ | ✘ |
| ICIS (Christensen et al., 2023) | ✔ | ✘ | ✔ | ✔ |
| LaBo (Yang et al., 2023) | ✘ | ✔ | ✔ | ✘ |
| ZSLViT (Chen et al., 2024) | ✘ | ✘ | ✔ | ✘ |
| CoMC (Liu et al., 2024) | ✘ | ✔ | ✔ | ✘ |
| DFZSL (Tang et al., 2024) | ✘ | ✘ | ✔ | ✘ |
| FuDD (Esfandiarpoor & Bach, 2024) | ✘ | ✘ | ✔ | ✘ |
| ASCI (Ours) | ✔ | ✔ | ✔ | ✔ |

**Discussion.** Existing I-ZSL methods rely on human-written class-level descriptions and fixed text encoders, which limit their capacity to distinguish fine-grained categories and adapt to domain-specific tasks. ASCI removes both constraints by generating task-oriented pairwise affinity descriptions using LLMs and by training a lightweight, task-adaptive semantic encoder jointly with the classifier injection process. This design enables stronger semantic alignment and improved generalization, without requiring visual data or manual supervision. Table 1 summarizes the modeling choices of recent approaches and shows that ASCI is the only method that simultaneously satisfies all three I-ZSL requirements: *no visual data*, *no human descriptions*, and *adaptive semantic guidance*. A more detailed discussion is provided in Appendix C.

## 3 Proposed Framework: ASCI

### 3.1 Preliminary

Let $\Phi : \mathbf{X} \rightarrow \mathcal{S}$ be a pre-trained classification model, where $\mathbf{X}$ represents the input feature space, and $\mathcal{S}$ is the set of *seen* class labels. $\Phi$ consists of two components: a feature extractor $\mathcal{F}$, which maps input features into a latent space, and a classifier with a weight matrix $\mathbf{W}_{\mathcal{S}}$ corresponding to the seen classes. The model $\Phi$ is pre-trained on a large-scale dataset labeled for the classes in $\mathcal{S}$.

**Image-Free Zero-Shot Learning.** The goal of I-ZSL is to extend $\Phi$ to classify instances from a target class set $\mathcal{Y}$ *without access to the visual data* $\mathbf{X}$. The extended model, denoted by $\hat{\Phi} : \mathbf{X} \rightarrow \mathcal{Y}$, must function in two settings: (*i*) *Zero-Shot Learning* (ZSL), where $\mathcal{Y} \cap \mathcal{S} = \emptyset$, meaning only unseen classes are classified; and (*ii*) *Generalized Zero-Shot Learning* (GZSL), where $\mathcal{Y}$ includes both seen and unseen classes, *i.e.*, $\mathcal{Y} \cap \mathcal{S} = \mathcal{S}$. The set of unseen classes is denoted as $\mathcal{U}$, where $\mathcal{U} = \mathcal{Y} \setminus \mathcal{S}$.

To extend $\Phi$ for classifying $\mathcal{U}$, we utilize semantic representations $\mathbf{A}$ that encode information about both seen classes $\mathcal{S}$ and unseen classes $\mathcal{U}$. These semantic representations are typically derived from predefined class descriptions, such as textual attributes or labels, which remain static and often require domain expertise to construct. Additionally, these predefined class-level descriptions are not optimal in capturing differences between classes, which is essential in I-ZSL. Moreover, the text encoders used to process $\mathbf{A}$ are typically trained on general-purpose corpora, making them less effective at capturing fine-grained, task-specific semantics. The discrepancy between these semantic representations and the classification objective can lead to degraded performance, particularly in domains with specialized requirements (Wang et al., 2019).

### 3.2 Overview of the Framework

The proposed framework, ASCI, addresses the challenges of I-ZSL by leveraging task-specific semantic relationships to dynamically inject classifiers for unseen classes. Unlike prior methods that rely on static predefined descriptions, ASCI generates adaptive semantic information, enabling effective generalization to unseen classes without requiring access to visual data features. As illustrated in Figure 1, ASCI comprises three key components: (*i*) a *class-pair affinity description generation module* establishes task-specific semantic relationships between seen and unseen classes using an LLM; (*ii*) an *adaptive semantic encoder* refines these relationships by projecting them into the semantic space of the pre-trained model; and (*iii*) a *classifier injection process* constructs classifiers for unseen classes based on the refined semantic representations and integrates them into the

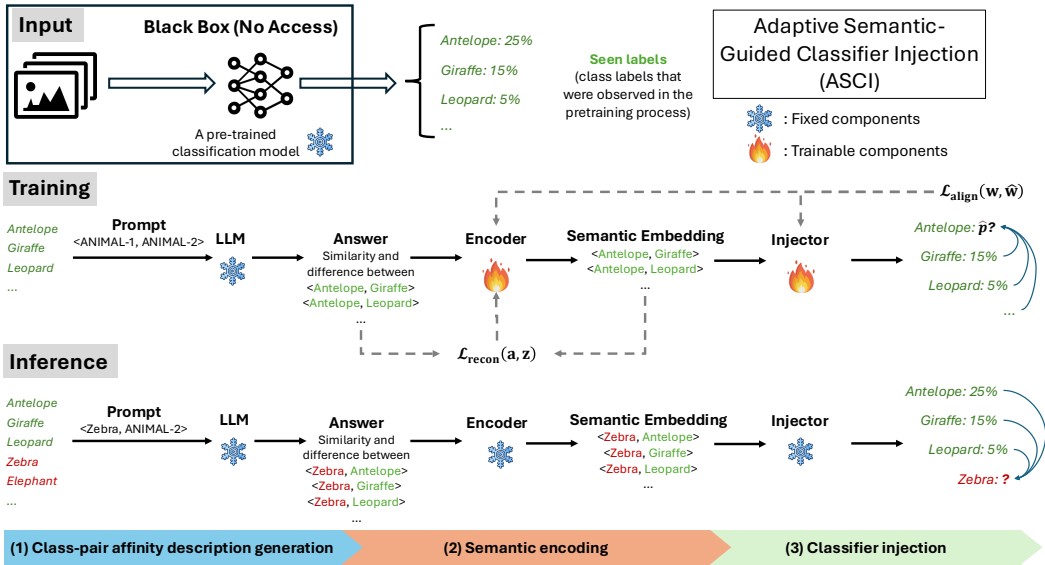

Figure 1: Overview of the ASCI framework.

extended model $\hat{\Phi}$. These components operate sequentially to extend the pre-trained model $\Phi$ for classifying unseen classes $\mathcal{U}$ while maintaining performance on seen classes $\mathcal{S}$.

### 3.3 CLASS-PAIR AFFINITY DESCRIPTION

Class-pair affinity descriptions capture the semantic relationships between seen and unseen classes, forming the foundation for generating classifiers for unseen classes. Instead of relying on static, predefined descriptions that require extensive manual curation and domain expertise, ASCI dynamically generates task-aware affinity descriptions using an LLM. This approach enhances adaptability across different classification tasks without requiring manually annotated semantic attributes.

Given a pair of classes $(c_i, c_j)$, where $c_i \in \mathcal{S}$ and $c_j \in \mathcal{U}$, the LLM generates a structured textual description that characterizes their semantic relationship. The prompt used to obtain these descriptions is provided in Appendix A.1. For example, in the CUB dataset (Wah et al., 2011), the relationship between a "sparrow" (*seen* class) and a "warbler" (*unseen* class) may be expressed as: "Both are small songbirds with similar feeding habits, but they differ in plumage coloration." These descriptions encapsulate class similarities and distinctions relevant to the classification task, aiding in the model's ability to infer relationships between seen and unseen classes.

To integrate these descriptions into the framework, they are first converted into numerical representations using a pre-trained text encoder. Let $f_{\text{LLM}}(c_i, c_j)$ denote the textual relationship generated by the LLM for the class pair $(c_i, c_j)$, and let $f_{\text{ENC}}$ represent the text encoder. The affinity embedding for the class pair is then computed as:

$$\mathbf{a}_{ij} = f_{\text{ENC}}(f_{\text{LLM}}(c_i, c_j)), \tag{1}$$

where $\mathbf{a}_{ij} \in \mathbb{R}^d$ represents the semantic embedding in a $d$-dimensional space.

The affinity embeddings for all class pairs are aggregated into an affinity matrix $\mathbf{A} \in \mathbb{R}^{|\mathcal{Y}| \times (|\mathcal{Y}|-1) \times d}$. This structured representation encodes relational information between classes and serves as input to the adaptive semantic encoder, which further processes and refines the embeddings to ensure alignment with the classification space.

### 3.4 ADAPTIVE SEMANTIC ENCODING FOR CLASSIFIER INJECTION

The adaptive semantic encoder refines class-pair affinity embeddings $\mathbf{a}_{ij}$ to align them with the classifier weights of the pre-trained model $\Phi$, enabling effective generalization to unseen classes. Under the I-ZSL setting, the feature extractor $\mathcal{F}$ is inaccessible, and the only available model outputs

are the classifier weight matrix for seen classes, $\mathbf{W}_{\mathcal{S}}$. Classifier weights for unseen classes are synthesized solely from semantic embeddings without relying on direct feature extraction.

**Semantic Embedding.** Given a class-pair semantic embedding $\mathbf{a}_{ij}$, the semantic encoder, parameterized by $\Theta$, maps it into a $k$-dimensional semantic space: $\mathbf{z}_{ij} = \Theta(\mathbf{a}_{ij})$, where $\mathbf{z}_{ij} \in \mathbb{R}^k$ represents the refined semantic representation, structured to align with the classifier weight space. To construct the classifier weight vector for an unseen class $c_j \in \mathcal{U}$, ASCI synthesizes $\mathbf{w}_j$ by adapting the classifier weights of seen classes based on their semantic relationships with $c_j$. Instead of learning new classifiers explicitly, the model generates $\mathbf{w}_j$ as a weighted combination of the classifier weights of seen classes:

$$\mathbf{w}_j = \sum_{c_i \in \mathcal{S}} \alpha_{ij} \cdot \mathbf{w}_i, \tag{2}$$

where $\mathbf{w}_i \in \mathbf{W}_{\mathcal{S}}$ represents the classifier weight of a seen class $c_i$, and $\alpha_{ij}$ quantifies the relationship between $c_i$ and $c_j$. The relationship weight is computed using softmax normalization:

$$\alpha_{ij} = \frac{\exp(f(\mathbf{z}_{ij} \oplus \mathbf{w}_i))}{\sum_{c_k \in \mathcal{S}} \exp(f(\mathbf{z}_{ik} \oplus \mathbf{w}_k))}. \tag{3}$$

Here, $\oplus$ denotes the concatenation operator, and $f(\cdot)$ is a function that measures the compatibility between the encoded semantic representation $\mathbf{z}_{ij}$ and the classifier weight $\mathbf{w}_i$. In this work, we implement $f(\cdot)$ using a multilayer perceptron (Gardner & Dorling, 1998).

By aligning the synthesized classifier weights with those of seen classes, this approach ensures that the classifier $\mathbf{w}_j$ for an *unseen* class remains semantically consistent with the underlying structure of $\Phi$. This alignment allows the model to infer classifiers for unseen categories without requiring image features or modifications to the pre-trained model.

**Classifier Injection and Model Extension.** The classifier injection module integrates the synthesized unseen-class classifiers $\mathbf{w}_j$ into the extended model $\hat{\Phi}$, enabling it to recognize both seen and unseen classes without additional retraining. It extends the pre-trained model $\Phi$ while preserving its original classification structure. The classifier weight matrix for the extended model is:

$$\mathbf{W}_{\mathcal{Y}} = \{\mathbf{W}_{\mathcal{S}}, \mathbf{W}_{\mathcal{U}}\}, \tag{4}$$

where $\mathbf{W}_{\mathcal{S}}$ consists of the pre-trained classifier weights for seen classes, and $\mathbf{W}_{\mathcal{U}}$ contains the synthesized classifier weights for unseen classes. Since the feature extractor is inaccessible, classification operates entirely within the semantic space. Instead of computing logits from extracted features, the unseen class weights $\mathbf{W}_{\mathcal{U}}$ are derived from their semantic relationship with $\mathbf{W}_{\mathcal{S}}$, ensuring that unseen classifiers maintain structural consistency with the pre-trained classification space.

### 3.5 Training and Inference

**Training.** To ensure that the adaptive semantic encoder $\Theta$ effectively projects class-pair affinity descriptions into the classification space, we introduce a training strategy that simulates the unseen-class scenario using only the seen classes. Specifically, for each class $c_j \in \mathcal{S}$, we temporarily mask its classifier weight $\mathbf{w}_j$ and reconstruct it using the remaining seen-class weights $\{\mathbf{w}_i | c_i \in \mathcal{S} \backslash \{c_j\}\}$. This self-supervised approach encourages the encoder to learn transferable representations that generalize to truly unseen classes.

The predicted weight for the masked class $c_j$ is obtained as: $\mathbf{w}_j = \sum_{c_i \in \mathcal{S} \backslash \{c_j\}} \alpha_{ij} \cdot \mathbf{w}_i$, where $\alpha_{ij}$ is computed via softmax normalization:

$$\alpha_{ij} = \frac{\exp(f(\mathbf{z}_{ij} \oplus \mathbf{w}_i))}{\sum_{c_k \in \mathcal{S} \backslash \{c_j\}} \exp(f(\mathbf{z}_{ik} \oplus \mathbf{w}_k))}. \tag{5}$$

To enforce consistency between the predicted and true classifier, we define an *alignment loss*:

$$\mathcal{L}_{\text{align}} = \sum_{c_j \in \mathcal{S}} \|\mathbf{w}_j - \sum_{c_i \in \mathcal{S} \backslash \{c_j\}} \alpha_{ij} \cdot \mathbf{w}_i\|_2^2. \tag{6}$$

This loss function encourages the model to generate classifier weights that maintain semantic consistency with $\Phi$. Additionally, a *reconstruction loss* is introduced to ensure that the encoded affinity embeddings retain the semantic relationships from the original class-pair descriptions. Specifically, the reconstruction loss penalizes deviations between the affinity embeddings $\mathbf{z}_{ij}$ and their original descriptions $\mathbf{a}_{ij}$ when decoded by an inverse mapping function $\Theta^{-1}$:

$$\mathcal{L}_{\text{recon}} = \sum_{c_i \in \mathcal{S}, c_j \in \mathcal{U}} \|\mathbf{a}_{ij} - \Theta^{-1}(\mathbf{z}_{ij})\|_2^2. \tag{7}$$

Here, $\Theta^{-1}$ is a decoder that reconstructs the original affinity descriptions, ensuring that the learned embeddings remain faithful to their semantic meaning. The total training objective is defined as:

$$\mathcal{L}_{\text{total}} = \mathcal{L}_{\text{align}} + \lambda \mathcal{L}_{\text{recon}}, \tag{8}$$

where $\lambda$ is a hyperparameter balancing classifier alignment and semantic reconstruction.

**Inference.** During inference, the extended model $\hat{\Phi}$ utilizes the generated classifier weights for unseen classes $\mathbf{W}_{\mathcal{U}}$ alongside the pre-trained classifier weights for seen classes $\mathbf{W}_{\mathcal{S}}$. Since the feature extractor $\mathcal{F}$ is inaccessible, classification is performed entirely in the semantic space by leveraging the alignment between the semantic embeddings $\mathbf{Z}$ and the classifier weights $\mathbf{W}$.

For a given query instance, the logits for both seen and unseen classes are computed as: $f_{\mathcal{S}} = \mathbf{W}_{\mathcal{S}}^{\top}\mathbf{Z}$, $f_{\mathcal{U}} = \mathbf{W}_{\mathcal{U}}^{\top}\mathbf{Z}$, where $\mathbf{Z}$ represents the semantic embeddings derived from the class-pair affinity descriptions. The final prediction is obtained by selecting the class with the highest score: $y^* = \arg\max_{y \in \mathcal{Y}} (f_{\mathcal{S}} + f_{\mathcal{U}})$. This inference strategy enables ASCI to classify unseen instances effectively without requiring raw data features, ensuring that predictions remain consistent with the learned semantic relationships. By leveraging a self-supervised training mechanism and performing inference directly in the semantic space, ASCI achieves robust generalization to unseen classes while preserving alignment with the pre-trained classifier weights. This approach enables efficient adaptation in both ZSL and GZSL settings without requiring additional fine-tuning.

## 3.6 THEORETICAL ANALYSIS

This section provides formal guarantees for the correctness and generalization capabilities of the proposed framework. We present a lemma to establish the alignment of the adaptive semantic encoding with the classifier weights of the pre-trained model and a theorem to bound the generalization error of the injected classifiers for unseen classes.

**Lemma 1.** *Let $\mathbf{a}_{ij}$ be the affinity embedding for the class pair $(c_i, c_j)$, where $c_i \in \mathcal{S}$ and $c_j \in \mathcal{U}$. If the adaptive semantic encoder $\Theta$ minimizes the alignment loss $\mathcal{L}_{align}$ as defined in Equation 6, then the transformed embedding $\mathbf{z}_{ij} = \Theta(\mathbf{a}_{ij})$ aligns with the classifier weights $\mathbf{W}_{\mathcal{S}}$ such that:*

$$\|\mathbf{w}_j^{true} - \mathbf{w}_j^{pred}\|_2^2 \leq \mathcal{O}(\mathcal{L}_{align}), \tag{9}$$

*where $\mathbf{w}_j^{true}$ is the ideal classifier weight, and $\mathbf{w}_j^{pred}$ is the predicted classifier weight synthesized from the affinity-based representations. The bound holds up to an approximation error $\epsilon$ due to the finite expressivity of $\Theta$.*

**Theorem 1.** *Let $\hat{\Phi}$ denote the extended model incorporating synthesized classifier weights $\mathbf{W}_{\mathcal{U}}$ for unseen classes. Suppose $\Theta$ is optimized to minimize the total loss $\mathcal{L}_{total}$ as defined in Equation 8. Under the assumption that the distributional shift between seen and unseen classes is bounded, the classification accuracy of $\hat{\Phi}$ on the target class set $\mathcal{Y}$ satisfies:*

$$Accuracy(\hat{\Phi}) \geq Accuracy(\Phi) - \mathcal{O}(\lambda\epsilon + \delta), \tag{10}$$

*where $\lambda$ is a trade-off parameter controlling the influence of the reconstruction loss, $\epsilon$ is the alignment error bound from Lemma 1, and $\delta$ represents the semantic variance of unseen classes.*

**Implications:** Lemma 1 and Theorem 1 provide theoretical justification for the framework's effectiveness. The adaptive semantic encoder $\Theta$ ensures that unseen-class classifier weights remain structurally consistent with the pre-trained model up to a bounded approximation error. Furthermore, the classification accuracy of the extended model $\hat{\Phi}$ degrades at most by $\mathcal{O}(\lambda\epsilon + \delta)$, ensuring that the generalization error remains controlled as long as $\mathcal{L}_{\text{align}}$ and $\mathcal{L}_{\text{recon}}$ are minimized effectively.

Table 2: Comparison between our ASCI framework and existing methods in the literature applicable or adaptable to the I-ZSL setting using standard benchmark (*i.e.*, CUB, AWA2, and SUN). We measure the results as unseen accuracy (Acc) for the zero-shot task, unseen (u) and seen (s) accuracy and their harmonic mean (H) for the generalised zero-shot setting. It reports the average number of 5 random runs with random seeds. Methods marked with * are adapted to the image-free setting.

| Models | Zero-Shot Accuracy | | | Generalised Zero-Shot Accuracy | | | | | | | | |
| | CUB Acc | AWA2 Acc | SUN Acc | CUB u | s | H | AWA2 u | s | H | SUN u | s | H |
|---|---|---|---|---|---|---|---|---|---|---|---|---|
| ConSE | $41.39_{\pm0.79}$ | $44.94_{\pm0.52}$ | $43.77_{\pm0.39}$ | $0.45_{\pm0.03}$ | $\mathbf{87.83}_{\pm0.87}$ | $0.90_{\pm0.04}$ | $3.22_{\pm0.13}$ | $\mathbf{96.29}_{\pm0.15}$ | $6.22_{\pm0.18}$ | $0.09_{\pm0.00}$ | $49.43_{\pm0.27}$ | $0.18_{\pm0.01}$ |
| COSTA | $33.62_{\pm1.21}$ | $46.30_{\pm1.63}$ | $18.68_{\pm0.97}$ | $0.00_{\pm0.00}$ | $87.82_{\pm1.70}$ | $0.00_{\pm0.00}$ | $0.00_{\pm0.00}$ | $96.27_{\pm1.72}$ | $0.00_{\pm0.00}$ | $0.00_{\pm0.00}$ | $\mathbf{51.90}_{\pm2.04}$ | $0.00_{\pm0.00}$ |
| Sub.Reg.* | $51.84_{\pm1.58}$ | $46.06_{\pm1.14}$ | $44.38_{\pm1.28}$ | $1.17_{\pm0.41}$ | $\mathbf{87.83}_{\pm1.98}$ | $2.30_{\pm0.13}$ | $0.00_{\pm0.00}$ | $96.26_{\pm1.25}$ | $0.00_{\pm0.00}$ | $0.00_{\pm0.00}$ | $51.55_{\pm1.50}$ | $0.00_{\pm0.00}$ |
| wDAE* | $48.88_{\pm0.90}$ | $52.47_{\pm1.30}$ | $45.76_{\pm2.53}$ | $0.93_{\pm0.05}$ | $87.45_{\pm1.44}$ | $1.85_{\pm0.09}$ | $0.00_{\pm0.00}$ | $95.86_{\pm2.17}$ | $0.00_{\pm0.00}$ | $0.00_{\pm0.00}$ | $50.43_{\pm1.04}$ | $0.00_{\pm0.00}$ |
| WAvg* | $2.00_{\pm0.43}$ | $20.42_{\pm0.61}$ | $1.39_{\pm0.53}$ | $1.75_{\pm0.20}$ | $58.97_{\pm2.67}$ | $3.40_{\pm0.49}$ | $8.77_{\pm0.10}$ | $87.05_{\pm2.06}$ | $15.93_{\pm1.11}$ | $1.34_{\pm0.35}$ | $7.57_{\pm0.90}$ | $2.28_{\pm0.18}$ |
| SMO* | $45.52_{\pm2.48}$ | $55.67_{\pm5.50}$ | $43.68_{\pm2.17}$ | $40.11_{\pm1.48}$ | $53.61_{\pm2.12}$ | $45.89_{\pm1.57}$ | $\mathbf{37.26}_{\pm0.82}$ | $86.44_{\pm2.18}$ | $\mathbf{52.08}_{\pm1.010}$ | $42.57_{\pm1.18}$ | $6.71_{\pm0.55}$ | $11.59_{\pm0.91}$ |
| ICIS | $63.43_{\pm1.22}$ | $21.65_{\pm0.11}$ | $52.14_{\pm1.37}$ | $52.21_{\pm1.60}$ | $70.07_{\pm1.63}$ | $59.83_{\pm1.72}$ | $0.01_{\pm0.00}$ | $96.24_{\pm1.93}$ | $0.01_{\pm0.00}$ | $42.11_{\pm0.74}$ | $28.56_{\pm1.16}$ | $34.03_{\pm1.18}$ |
| LaBo | $50.83_{\pm1.30}$ | $43.99_{\pm1.14}$ | $48.22_{\pm1.32}$ | $46.89_{\pm2.64}$ | $57.45_{\pm2.23}$ | $49.01_{\pm2.85}$ | $33.07_{\pm0.86}$ | $80.40_{\pm2.29}$ | $45.76_{\pm1.56}$ | $43.81_{\pm1.70}$ | $7.52_{\pm0.88}$ | $11.12_{\pm0.05}$ |
| ZSLViT* | $64.17_{\pm2.30}$ | $52.04_{\pm1.99}$ | $50.63_{\pm2.03}$ | $53.44_{\pm1.90}$ | $72.21_{\pm2.78}$ | $60.94_{\pm2.36}$ | $30.39_{\pm1.37}$ | $82.95_{\pm2.01}$ | $50.06_{\pm0.83}$ | $41.11_{\pm1.93}$ | $26.59_{\pm0.86}$ | $33.32_{\pm0.54}$ |
| CoMC* | $56.29_{\pm1.82}$ | $49.28_{\pm1.00}$ | $44.27_{\pm1.75}$ | $10.21_{\pm0.46}$ | $77.81_{\pm3.23}$ | $6.01_{\pm0.04}$ | $3.45_{\pm0.10}$ | $\mathbf{96.29}_{\pm1.42}$ | $7.56_{\pm0.57}$ | $36.40_{\pm1.76}$ | $25.51_{\pm0.87}$ | $31.40_{\pm0.57}$ |
| DFZSL* | $54.21_{\pm1.12}$ | $48.37_{\pm1.04}$ | $45.63_{\pm1.26}$ | $41.09_{\pm0.93}$ | $67.42_{\pm1.77}$ | $51.08_{\pm1.21}$ | $28.33_{\pm0.72}$ | $83.95_{\pm2.01}$ | $42.27_{\pm0.84}$ | $33.71_{\pm0.66}$ | $25.94_{\pm0.92}$ | $29.23_{\pm0.74}$ |
| FuDD | $43.62_{\pm1.32}$ | $52.55_{\pm1.97}$ | $40.14_{\pm0.44}$ | $43.39_{\pm1.20}$ | $51.92_{\pm1.14}$ | $41.24_{\pm1.32}$ | $36.29_{\pm0.85}$ | $88.82_{\pm1.73}$ | $47.62_{\pm1.38}$ | $39.92_{\pm0.92}$ | $5.00_{\pm0.00}$ | $2.11_{\pm0.01}$ |
| ASCI (Ours) | $\mathbf{70.39}_{\pm0.64}$ | $\mathbf{59.92}_{\pm0.93}$ | $\mathbf{58.28}_{\pm0.87}$ | $\mathbf{64.29}_{\pm0.42}$ | $79.93_{\pm0.37}$ | $\mathbf{70.62}_{\pm0.77}$ | $36.40_{\pm0.33}$ | $\mathbf{96.29}_{\pm0.29}$ | $51.39_{\pm0.71}$ | $\mathbf{47.76}_{\pm0.22}$ | $44.77_{\pm0.82}$ | $\mathbf{41.53}_{\pm0.74}$ |

Scalability is a key consideration in zero-shot learning settings. As the number of unseen classes $|\mathcal{U}|$ increases, the effect of unseen-class variance $\delta$ grows, potentially impacting performance. This requires careful tuning of $\lambda$ to balance classifier alignment and generalization (relevant experimental investigations can be found in Section 4.3). These findings establish that the proposed method preserves semantic alignment and ensures effective generalization in both ZSL and GZSL scenarios. Detailed proofs and further discussions are provided in Appendix D.

# 4 EXPERIMENTS

## 4.1 DATASETS AND COMPETING MODELS

We evaluate the I-ZSL performance on three widely used benchmark datasets, covering both fine-grained and coarse-grained classification tasks, including CUB (Wah et al., 2011), AWA2 (Xian et al., 2019), and SUN (Patterson et al., 2014). To ensure consistency across experiments, we adopt the class splits from (Christensen et al., 2023) for ZSL and GZSL. A detailed description of these datasets is provided in Appendix E.1. We compare ASCI against ZSL and I-ZSL methods, *e.g.*, ConSE (Norouzi et al., 2014), COSTA (Mensink et al., 2014), Sub.Reg.* (Akyürek et al., 2022), wDAE* (Gidaris & Komodakis, 2019), WAvg* (Xu et al., 2022), SMO* (Xu et al., 2022), ICIS (Christensen et al., 2023), LaBo (Yang et al., 2023), ZSLViT* (Chen et al., 2024), CoMC* (Liu et al., 2024), DFZSL (Tang et al., 2024) and FuDD (Esfandiarpoor & Bach, 2024). *For methods marked with *, which require image features, we replace image inputs* $\mathbf{X}$ *with the weight matrix* $\mathbf{W}$ *to ensure a fair comparison under the I-ZSL paradigm.* Detailed descriptions are provided in Appendix E.2, with implementation available in our code.

## 4.2 MAIN RESULTS

**Zero-shot learning performance.** Table 2 compares ASCI with existing methods under the standard ZSL setting across CUB, AWA2, and SUN. ASCI achieves the highest zero-shot accuracy on all datasets, outperforming both image-free methods and those adapted to this setting. Notably, ASCI surpasses ICIS (Christensen et al., 2023), the previous best-performing I-ZSL method, by 7.0% on CUB and 6.9% on SUN, demonstrating the effectiveness of adaptive semantic representations. Compared to ZSLViT (Chen et al., 2024), which benefits from vision transformer-based feature aggregation, ASCI achieves superior accuracy despite operating in a fully image-free setting. These results highlight the advantage of class-pair affinity descriptions and adaptive text encoding in capturing meaningful semantic relationships between seen and unseen classes.

**Generalized zero-shot learning performance.** GZSL presents an additional challenge, requiring the model to distinguish between both seen and unseen classes. ASCI demonstrates strong generalization, achieving the highest harmonic mean (H) across all datasets. On CUB, ASCI achieves an H-score of 70.62%, surpassing ICIS by 11.6%. On AWA2, ASCI maintains a balance between unseen and seen class accuracy, achieving an H-score of 51.39%, while other methods struggle with

Table 3: Evaluation of the contribution of key components by removing the affinity description and semantic encoding modules, as well as by varying the hyperparameter $\lambda$. Without the affinity description module, we replace the class-pair generated descriptions with target class labels and class-wise descriptions. The semantic encoding module is removed to assess its role in aligning class representations with the classification space. We replace the generated class-pair descriptions with the concatenation of two class-level descriptions. The default setting uses $\lambda = 1$; we report results for $\lambda = 0.5$ (weaker semantic reconstruction) and $\lambda = 2$ (stronger reconstruction emphasis).

| Settings | Zero-Shot Accuracy | | | Generalised Zero-Shot Accuracy | | | | | | | | |
| | CUB Acc | AWA2 Acc | SUN Acc | CUB | | | AWA2 | | | SUN | | |
| | | | | u | s | H | u | s | H | u | s | H |
| ASCI (Ours) | $70.39_{\pm0.64}$ | $59.92_{\pm0.93}$ | $58.28_{\pm0.87}$ | $64.29_{\pm0.42}$ | $79.93_{\pm0.37}$ | $70.62_{\pm0.77}$ | $36.40_{\pm0.33}$ | $96.29_{\pm0.29}$ | $51.39_{\pm0.71}$ | $47.76_{\pm0.22}$ | $44.77_{\pm0.82}$ | $41.53_{\pm0.74}$ |
| w/o Affinity Description | $63.85_{\pm0.65}$ | $53.42_{\pm0.89}$ | $51.27_{\pm0.85}$ | $55.12_{\pm0.37}$ | $78.31_{\pm0.35}$ | $63.95_{\pm0.50}$ | $31.22_{\pm0.21}$ | $94.78_{\pm0.16}$ | $45.80_{\pm0.30}$ | $40.83_{\pm0.30}$ | $42.95_{\pm0.77}$ | $38.17_{\pm0.62}$ |
| w/ Class Description | $64.23_{\pm3.81}$ | $55.23_{\pm2.50}$ | $57.83_{\pm0.92}$ | $59.00_{\pm1.24}$ | $78.72_{\pm0.71}$ | $65.03_{\pm1.24}$ | $33.23_{\pm0.80}$ | $95.27_{\pm1.72}$ | $48.04_{\pm1.70}$ | $45.52_{\pm0.51}$ | $43.05_{\pm1.51}$ | $39.13_{\pm0.88}$ |
| w/o Semantic Encoding | $61.74_{\pm0.61}$ | $52.63_{\pm0.92}$ | $49.81_{\pm0.84}$ | $50.48_{\pm0.61}$ | $77.91_{\pm0.42}$ | $60.94_{\pm0.82}$ | $28.37_{\pm0.44}$ | $92.64_{\pm0.71}$ | $43.29_{\pm0.28}$ | $37.55_{\pm0.31}$ | $41.33_{\pm0.73}$ | $35.79_{\pm0.90}$ |
| $\lambda = 0.5$ | $67.89_{\pm0.52}$ | $57.71_{\pm0.88}$ | $55.36_{\pm0.58}$ | $60.41_{\pm1.06}$ | $79.26_{\pm0.94}$ | $67.50_{\pm0.99}$ | $34.61_{\pm0.30}$ | $95.11_{\pm0.93}$ | $49.54_{\pm0.39}$ | $45.92_{\pm0.20}$ | $43.68_{\pm0.39}$ | $39.21_{\pm0.41}$ |
| $\lambda = 2$ | $66.22_{\pm0.64}$ | $56.38_{\pm0.82}$ | $54.19_{\pm0.59}$ | $58.74_{\pm0.83}$ | $78.32_{\pm0.92}$ | $65.97_{\pm0.92}$ | $33.84_{\pm0.29}$ | $96.12_{\pm0.22}$ | $47.83_{\pm0.23}$ | $44.15_{\pm0.12}$ | $41.47_{\pm0.53}$ | $38.58_{\pm0.34}$ |
| Only LLM (GPT-4) | $11.37_{\pm0.45}$ | $47.82_{\pm0.62}$ | $26.94_{\pm0.57}$ | $8.42_{\pm0.38}$ | $62.15_{\pm0.70}$ | $14.88_{\pm0.41}$ | $41.26_{\pm0.59}$ | $69.37_{\pm0.65}$ | $51.73_{\pm0.52}$ | $18.73_{\pm0.44}$ | $47.62_{\pm0.61}$ | $26.52_{\pm0.48}$ |

overfitting to seen categories. This improvement is attributed to the dynamic classifier injection mechanism, which ensures that unseen class representations remain aligned with the pre-trained classification space while preserving performance on seen classes.

**Fine-grained recognition.** ZSL performance is sensitive to fine-grained distinctions, where inter-class similarities can make classification challenging. This issue is especially evident in CUB, which contains visually similar bird species. ASCI excels in such cases, improving unseen accuracy by 6.4% over ICIS. The adaptive semantic encoding mechanism enables ASCI to capture subtle differences between similar species, addressing the limitations of static, predefined semantic representations. Similarly, the performance gain on SUN, a scene classification dataset with semantically overlapping categories, further demonstrates the effectiveness of ASCI in distinguishing fine-grained classes.

### 4.3 FURTHER ANALYSIS

Due to space constraints, this section only provides a brief discussion. Detailed experimental settings and extended discussions of the analysis experiments are provided in Appendix F.

**Effect of removing class-pair affinity descriptions.** Table 3 shows that replacing class-pair affinity descriptions with either class labels or class-wise descriptions reduces generalization to unseen classes. Specifically, removing affinity descriptions leads to a 6.54% drop in zero-shot accuracy on CUB ($70.39\% \rightarrow 63.85\%$) and a 7.01% drop on SUN ($58.28\% \rightarrow 51.27\%$). The H-score on SUN also declines by 3.36% ( $41.53\% \rightarrow 38.17\%$), highlighting the importance of structured semantic relationships for fine-grained recognition. In addition, using class-wise descriptions performs better than class labels alone. For example, on CUB, using class-wise descriptions yields $64.23\%$ accuracy, compared to $63.85\%$ with class labels, showing that additional semantic content—though lacking explicit relational information—still improves generalization compared to label-only inputs.

**Effect of removing adaptive semantic encoding.** Removing the semantic encoding module leads to a notable decline in performance, with zero-shot accuracy dropping by 8.65% on CUB ($70.39\% \rightarrow 61.74\%$) and 8.47% on SUN ($58.28\% \rightarrow 49.81\%$). The harmonic mean (H) on AWA2 decreases by 9.60% ($51.39\% \rightarrow 41.79\%$), emphasizing the essential role of adaptive semantic encoding in aligning class representations with the classification space. These findings indicate that static text embeddings are insufficient and that adaptive encoding is necessary for effectively refining affinity descriptions and supporting generalization to unseen classes.

**Impact of $\lambda$.** The hyperparameter $\lambda$ determines the trade-off between semantic reconstruction and classifier alignment. When $\lambda = 0.5$, unseen class accuracy increases (+0.88% on CUB and +2.15% on AWA2), resulting in a higher H-score, but seen-class accuracy decreases slightly, likely due to reduced emphasis on classifier alignment. In contrast, setting $\lambda = 2$ increases seen-class accuracy (+1.06% on AWA2), but decreases unseen-class accuracy ($-5.55\%$ on SUN and $-4.85\%$ on AWA2), which leads to a lower H-score as a result of over-regularization. Overall, these results indicate that $\lambda = 1$ achieves the best balance, supporting adaptability to unseen classes while maintaining performance on seen classes.

**Using LLMs as standalone classifiers.** Table 3 further reports the results of directly using GPT-4 for classification by providing images and candidate class lists (prompt in Appendix A.3). While GPT-4 attains moderate performance on coarse-grained datasets such as AWA2 ($H = 51.73\%$), it performs poorly on fine-grained datasets, yielding only $14.88\%$ $H$ on CUB. In contrast, our method achieves consistent improvements across all three benchmarks, with $H = 70.62\%$ on CUB and $H = 51.39\%$ on AWA2. This comparison highlights the necessity of our adaptive semantic-guided design, as large language models alone are insufficient for reliable generalized zero-shot learning.

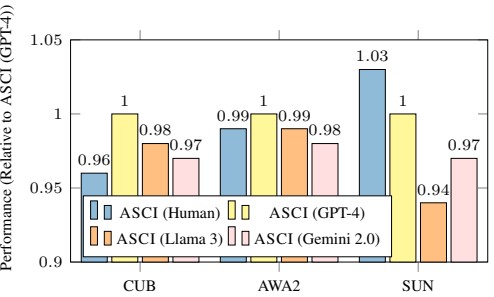 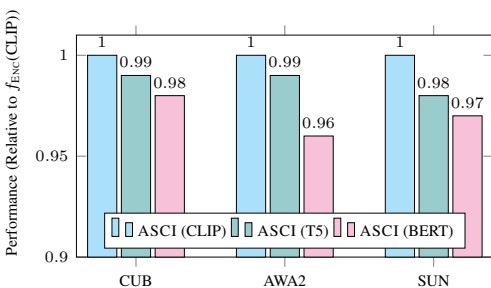

(a) Relative performance comparison of ASCI variants with different LLMs and human annotators.

(b) Relative performance comparison of ASCI variants with different text encoders ($f_{\text{ENC}}$).

Figure 2: Ablation study.

**Different description annotators.** Figure 2a compares ASCI performance using descriptions from humans and various LLMs. GPT-4 achieves the best results and serves as the baseline. Llama 3 performs comparably, with only a slight decrease across datasets, while Gemini 2.0 shows a more significant drop, especially on SUN. Human-generated descriptions yield strong but not top performance. This may be because the student annotators lack sufficient fine-grained domain knowledge to provide precise and consistent semantic distinctions between classes, especially in specialized or subtle cases. These results highlight that high-quality LLMs can generate class-pair descriptions that are at least as effective, if not better, than those from non-expert human annotators, particularly for tasks requiring detailed domain understanding. Examples are provided in Appendix B.

**Impact of different $f_{\text{ENC}}$.** The choice of text encoder ($f_{\text{ENC}}$) influences classification performance, as shown in Figure 2b. T5 (Raffel et al., 2020)-based embeddings achieve near-CLIP (Radford et al., 2021) performance across all datasets, with minor variations. BERT (Devlin et al., 2019)-based embeddings perform slightly worse, particularly on AWA2, where they reach $0.96$ relative to CLIP. This suggests that T5's sequence-to-sequence pretraining contributes to stronger contextual representations, while BERT's bidirectional training, though effective, does not generalize as well for this task. These findings emphasize the importance of selecting a text encoder that aligns well with the dataset's semantic structure to optimize performance.

**Can other methods benefit from LLM generated descriptions?** We substitute human-written descriptions in baseline methods with LLM-generated ones and present the results in Table 4. Comparison with Table 2 shows that most baselines exhibit little or no improvement, suggesting that simply using LLM-generated descriptions is not sufficient for better performance. In contrast, our model is explicitly designed to exploit the richer semantic information from LLM-generated descriptions and achieves clear gains, highlighting the need for adaptive semantic modeling in I-ZSL.

## 5 CONCLUDING REMARKS

This work introduces a novel framework designed for I-ZSL. Unlike previous methods that rely on pre-defined class descriptions and static text encoders, ASCI dynamically generates class-pair affinity descriptions using LLMs and employs an adaptive text encoder to refine semantic representations. By aligning classifier weights with task-specific semantics, ASCI enhances generalization to unseen classes while operating in fully image-free settings. Extensive experiments on standard benchmarks demonstrate that ASCI significantly improves I-ZSL performance, particularly in fine-grained classification tasks. A discussion of limitations and future work is provided in Appendix G.

ETHICS STATEMENT

All datasets used in this work (CUB, AWA2, SUN) are publicly available under terms that permit research use, and we strictly follow the official training/test splits. Our method does not involve private or sensitive data. While LLMs are employed to generate semantic descriptions, these are used solely for research purposes and do not contain personal or identifiable information. Potential risks such as bias or factual errors in LLM-generated text are mitigated by our adaptive semantic modeling, and all experiments are bounded within controlled benchmarks. We therefore do not identify direct pathways to harmful applications such as surveillance or privacy-invasive systems.

REPRODUCIBILITY STATEMENT

We have made our implementation and experimental settings publicly available at `https://anonymous.4open.science/r/ASCI-828D`. The repository includes data preprocessing scripts, training and evaluation pipelines, and instructions for reproducing all results. Experiments were conducted using PyTorch 2.2 with CUDA 12.1 on NVIDIA A100 GPUs. Detailed hyperparameters and training schedules are listed in Appendix E. We fix random seeds and report mean and standard deviation over multiple runs. Together, these steps ensure that all results in this paper can be reproduced by independent researchers.

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

## A  PROMPT

### A.1  CLASS-PAIR AFFINITY DESCRIPTION GENERATION PROMPT

---

*Class-pair affinity description generation prompt*

**Instruction:** You are tasked with comparing two animal species based on their semantic relationships. Use your knowledge to generate a detailed and accurate description of the similarities and differences between the given animals. Ensure the comparison is domain-relevant and includes key characteristics that would help identify or distinguish these species in a real-world classification task.

**Message:** Describe the similarities and differences between [ANIMAL-1] and [ANIMAL-2]. Focus on key aspects such as their physical characteristics, habitat, diet, behavior, and any notable distinctions in their roles in the ecosystem. Provide a concise but detailed explanation that highlights how these animals are alike and how they differ.
**Response:** [The response will contain a structured and precise description of the relationship between the two animals, highlighting shared features and distinct attributes relevant to their classification.]

---

### A.2  CLASS-WISE DESCRIPTION GENERATION PROMPT

---

*Class-wise description generation prompt*

**Instruction:** You are tasked with describing one animal species based on its characteristics. Use your knowledge to generate a detailed and accurate description of the given animal. Ensure the description is domain-relevant and includes key characteristics that would help identify this species in a real-world classification task.

**Message:** Describe the [ANIMAL]. Focus on key aspects such as its physical characteristics, habitat, diet, behavior, and any notable distinctions in its roles in the ecosystem.
**Response:** [The response will contain a structured and precise description of the given animal.]

---

### A.3  PROMPT FOR USING LLMS AS CLASSIFIERS

---

*Prompt for using LLMs as classifiers*

**Instruction:** You are tasked with classifying an image into one of the given candidate categories. You are provided with the input image and a candidate set of class labels. Select the most appropriate class from this candidate set based on the visual content of the image. Provide only the final predicted class label without additional explanation.

**Message:** Input image: [IMAGE]
Candidate class labels: [CLASS-1, CLASS-2, ..., CLASS-N]
Which class does the image most likely belong to? Provide only the class label as the final answer.

**Response:** Predicted class: [CLASS-X]

---

## B  EXAMPLE GENERATED DESCRIPTIONS

This section presents the example generated descriptions from LLMs (GPT-4, Llama 3, and Gemini 2.0) and human annotators. Regarding human annotators, we ask 3 bachelor students with a computer science or mathematics background to perform this task.

**CUB.**  *Coot vs. Grebe*

- *Human:* Coots and Grebes are both aquatic birds found in lakes and ponds, and both have lobed toes for swimming. Coots are dark-colored with a white beak and often feed on plants and small animals at the water's edge. Grebes have sleeker bodies, pointed bills, and are expert divers that mainly eat fish and insects. While both occupy similar habitats, coots

are more likely to be seen walking on land, whereas grebes spend more time diving and rarely leave the water.

- *GPT-4:* Coot and Grebe are both freshwater diving birds with lobed feet and similar diets. Coots have a stocky body, blunt beak, and walk well on land. Grebes are sleek, have sharp beaks, and are clumsy on land. Coots are social; Grebes are more solitary and perform courtship displays.

- *Llama 3:* The Coot and the Grebe are two distinct waterbird species. Similarities include their aquatic lifestyle, freshwater habitats, and diet of aquatic plants and invertebrates.

  Differences lie in their physical characteristics, with Coots having a robust body, black plumage, and a white bill, whereas Grebes have a sleeker body and grayish or brownish plumage. Grebes are excellent divers, preferring open water, while Coots are more surface-oriented, favoring vegetated areas.

  Their behaviors also differ, with Grebes being more solitary during breeding and Coots forming larger groups. In the ecosystem, both species play important roles, but Coots have a greater impact on vegetation due to their plant-based diet. Overall, while sharing some similarities, the Coot and the Grebe exhibit distinct adaptations to their environments.

- *Gemini 2.0:* Similarities: Aquatic birds, swim/dive, eat aquatic invertebrates/some plants, build floating nests.

  Differences: Coots: Dark plumage, white bill & frontal shield, lobed toes (can walk on land), omnivorous, often in flocks. Grebes: Streamlined, pointed bill, distinctly lobed feet (poor walking), mainly carnivorous (fish/invertebrates), often solitary/pairs. Key Distinctions: Bill/foot morphology and terrestrial ability.

**CUB.** *Wren vs. Finch*

- *Human:* Wrens and Finches are both small songbirds commonly found in gardens and forests. Wrens have thin, slightly curved bills, short wings, and are known for their upright tails and insect-based diet. Finches have stout, conical bills adapted for eating seeds, and generally display a more robust body shape. While both may sing and nest in shrubs, wrens are more active in searching for insects, while finches mainly forage for seeds on the ground or in trees.

- *GPT-4:* Wren and Finch are both small songbirds often found in gardens and woodlands. Wrens are brown, with short wings and tails held upright; they eat insects and are very active. Finches are more colorful, with conical beaks for seeds, and show more varied plumage. Wrens are solitary; Finches often form flocks.

- *Llama 3:* The Wren and the Finch are two distinct bird species. Similarities include their small size, insectivorous and seed-eating diets, and social behaviors, often forming flocks.

  Differences lie in their physical characteristics, with Wrens having a rounder body, shorter tail, and longer bill, whereas Finches have a more elongated body, notched tail, and shorter bill. Wrens are also known for their bold and loud songs, whereas Finches have more melodic and varied songs.

  In terms of habitat, Wrens prefer denser vegetation, such as shrubs and trees, while Finches are found in a wider range of habitats, including backyards, fields, and woodlands. Overall, while sharing some similarities, the Wren and the Finch exhibit distinct characteristics and preferences.

- *Gemini 2.0:* Similarities: Small passerine birds, diurnal, known for songs.

  Differences: Wrens: Tiny, roundish, thin bill, short tail (often upright), mainly brown, primarily insectivorous, often solitary, dome-shaped nests. Finches: Slightly larger, stout, conical bill, varied tail, often colorful, primarily granivorous, sometimes in flocks, cup-shaped nests. Key Distinctions: Bill shape (thin vs. conical) and primary diet (insects vs. seeds).

**AWA2.** *Wolf vs. Domestic Dog*

- *Human:* Wolves and domestic dogs are closely related canines, both with similar body structures and social behaviors. Wolves are wild animals with larger bodies, longer legs,

and thick, grayish fur; they live and hunt in packs and rely on hunting large prey in forests or open lands. Domestic dogs are smaller on average, have diverse fur colors and shapes due to selective breeding, and live with humans, eating a variety of foods. Unlike wolves, dogs rely on humans for care and rarely hunt in packs.

- *GPT-4:* Wolf and Domestic Dog share ancestry and body plan. Wolves are larger, have broader skulls, and stronger jaws. They live in wild packs and hunt large prey. Dogs vary in size and shape due to breeding, live with humans, and depend on them for food. Wolves avoid humans; dogs are domesticated and socialized.

- *Llama 3:* The Wolf and the Domestic Dog are two closely related species. Similarities include their carnivorous diet, pack behavior, and loyal nature.

  Differences lie in their physical characteristics, with Wolves having a more muscular build, sharper teeth, and a thicker coat, whereas Domestic Dogs have a more varied size, shape, and coat type due to breeding. Wolves are also more territorial and have a stronger prey drive, whereas Domestic Dogs are more socialized to humans and have a wider range of behaviors and roles, from companions to working animals. Overall, while sharing a common ancestor, Wolves and Domestic Dogs have distinct differences in their physiology, behavior, and relationship with humans.

- *Gemini 2.0:* Similarities: Same species (Canis lupus), pack-oriented, quadrupedal carnivores, similar vocalizations.

  Differences: Wolves: Larger, powerful build, straighter tail, muted colors, more cautious/independent, shorter socialization. Domestic Dogs: Wide size/shape range, varied tail, diverse colors, more amenable to humans, longer socialization. Key Distinctions: Physical variability/size, temperament/dependence on humans.

**AWA2.** *Horse vs. Zebra*

- *Human:* Horses and zebras are both hoofed mammals in the Equidae family. Horses have solid-colored coats, are domesticated, and are used for riding, transport, or farm work. Zebras are wild animals, easily identified by their black-and-white stripes, and live in herds on African grasslands. While both mainly eat grasses, horses are more adaptable to different environments, while zebras rely on their stripes for camouflage and social behavior in the wild.

- *GPT-4:* Horse and Zebra are both equids with similar body shape and herbivorous diet. Horses are domesticated, come in various colors, and are used for riding and work. Zebras are wild, have black-and-white stripes, and are more aggressive and harder to tame. Horses neigh and are social with humans; zebras bark and live in tight herds for predator defense.

- *Llama 3:* The Horse and the Zebra are two closely related species. Similarities include their herbivorous diet, social behavior, and grazing habits.

  Differences lie in their physical characteristics, with Horses having a solid coat color and Zebras having distinctive black and white stripes. Zebras are also more adapted to wilderness habitats and have a stronger instinct for self-preservation, whereas Horses are more domesticated and have a wider range of behaviors and roles, from companions to working animals. Overall, while sharing a common ancestor, Horses and Zebras have distinct differences in their physiology, behavior, and relationship with their environment.

- *Gemini 2.0:* Similarities: Equids, herbivorous, similar body plan (hooves, legs, tail, mane), live in herds.

  Differences: Zebras: Black and white stripes, stockier build, coarser mane, less domesticable, native to Africa. Horses: Typically solid-colored coats (or spots), more varied build, often longer mane, generally domesticable, global distribution (wild populations more restricted). Key Distinction: Coat pattern (stripes vs. solid/spots).

**SUN.** *Classroom vs. Library*

- *Human:* Classrooms and libraries are both indoor environments designed for learning. Classrooms typically contain desks, a teacher's area, and boards for instruction, and are used for active teaching and group discussions. Libraries are quieter spaces filled with

bookshelves, reading tables, and study areas, intended for reading, research, and independent study. While both support education, classrooms focus on direct instruction, while libraries provide resources and a quiet atmosphere for individual learning.

- *GPT-4:* Classroom and Library are both indoor educational spaces. A classroom is for instruction, with a teacher, desks, and active discussion. A library is for independent study, with books, quiet zones, and minimal interaction. Classrooms focus on teaching; libraries support learning through resources.

- *Llama 3:* The Horse and the Zebra are closely related equine species. Similarities include their herbivorous diet, four-legged stance, and social behavior, often living in groups.

  Differences lie in their physical characteristics, with Horses having a solid coat color and a more muscular build, whereas Zebras have a distinctive black and white striped pattern and a slightly smaller size. Zebras also have a more aggressive temperament and are native to Africa, whereas Horses are found in various parts of the world and have been domesticated for thousands of years. Overall, while sharing many similarities, the Horse and the Zebra exhibit distinct differences in their appearance, behavior, and geographic distribution.

- *Gemini 2.0:* Similarities: Spaces for learning, contain informational resources, found in educational settings.

  Differences: Classroom: For structured teaching with an instructor, curriculum-focused, scheduled attendance. Library: For self-directed learning, diverse resources, voluntary access. Key Distinction: Presence of a teacher/structured instruction vs. self-directed resource access.

**SUN.** *Beach vs. Desert*

- *Human:* Beaches and deserts are both natural environments with sandy ground. Beaches are located next to bodies of water like oceans or lakes, have moist sand, support aquatic life, and are often visited for recreation. Deserts are dry, receive very little rainfall, have arid sand or rocky soil, and support sparse, drought-resistant vegetation and animals. While both can be hot and sunny, beaches are shaped by water and tides, while deserts are shaped by wind and extreme dryness.

- *GPT-4:* Beach and Desert are both sandy environments but differ in key aspects. Beaches are coastal, humid, and support marine life; deserts are inland, arid, and support drought-adapted species. Beaches have tides and moderate temperatures; deserts have extreme temperature swings and minimal water.

- *Llama 3:* The Beach and the Desert are two distinct ecosystems. Similarities include their ability to support unique and adapted plant and animal species.

  Differences lie in their climate, geography, and characteristics, with Beaches being coastal areas with sandy or rocky shores, salty air, and a variety of marine life, whereas Deserts are arid, landlocked regions with extreme temperatures, limited vegetation, and specialized wildlife. Beaches are also typically characterized by high humidity and abundant water, whereas Deserts are defined by low humidity and scarce water resources. Overall, the Beach and the Desert exhibit contrasting environments and support different types of biodiversity.

- *Gemini 2.0:* Similarities: Sparse vegetation, large areas of sediment, can have extreme temperatures, subject to wind/water erosion.

  Differences: Beach: Adjacent to water, higher water availability/humidity, vegetation adapted to salt/waves, temperature moderated by water, shaped by water. Desert: Arid (low water), vegetation adapted to drought, extreme daily temperature swings, shaped by wind. Key Distinction: Presence/absence of significant water.

## C RELATED WORK DISCUSSION

Table 1 provides a side-by-side comparison of ASCI and existing approaches in image-free zero-shot learning (I-ZSL). We highlight three key dimensions relevant to I-ZSL: the need for manually defined class descriptions, the use of adaptive semantic guidance, and the reliance on image vision features.

Most early methods, such as ConSE (Norouzi et al., 2014) and COSTA (Mensink et al., 2014), support I-ZSL but require human-written class descriptions and do not employ adaptive semantic modeling. Methods like wDAE (Xian et al., 2017) and WAvg (Socher et al., 2013) do not support I-ZSL at all, since they depend on visual features for both training and inference. ICIS (Christensen et al., 2023) is a representative I-ZSL model that removes the dependency on images but still relies on manually defined class descriptions and a fixed semantic encoder, without adaptive guidance for semantic representation.

In contrast, ASCI is the first to achieve the following:

- No reliance on human-written class descriptions: All semantic information is generated automatically by LLMs, removing the need for expert annotation and enabling adaptation to new domains where such information is unavailable or difficult to collect.

- Adaptive semantic guidance: Our model introduces a task-adaptive encoder trained jointly with the classifier, rather than relying on a fixed pre-trained text encoder. This design allows the semantic space to be optimized for the target task, improving discrimination between fine-grained classes.

- Fully image-free: ASCI does not require any image data or models at any stage, making it applicable to sensitive domains where visual data or models are inaccessible due to privacy, security, or resource constraints.

Compared to prior methods, our approach is the only one to satisfy all three criteria: operating fully in the image-free setting, without human-crafted class descriptions, and with adaptive semantic modeling. This enables a more flexible and robust framework for I-ZSL, as summarized in Table 1. Our results demonstrate that these design choices translate to clear improvements in both accuracy and generalization, particularly in fine-grained or specialized domains where existing assumptions are not realistic.

**Costs of LLM Usages.** Our automatic class-pair affinity description generation pipeline issued a total of 137 queries to GPT-4 and 135 queries to Gemini 2.0. Each request used a prompt of approximately 150 tokens and received a model response of about 100 tokens, for a total of roughly 250 tokens per request. Based on research-tier pricing ($0.06 per 1K prompt tokens and $0.12 per 1K completion tokens for GPT-4; $0.02 per 1K prompt/completion tokens for Gemini 2.0), the average cost per GPT-4 request was approximately $0.021, and per Gemini 2.0 request was about $0.005. Summing across all calls, the total expenditure was about $2.88 for GPT-4 and $0.68 for Gemini 2.0, for a combined API cost of less than $4. This total is significantly lower than previous estimates based on 1K+1K token usage. Most costs are attributable to GPT-4, which performs the majority of description generation, while Gemini 2.0 is used for ablation studies. Local experiments with Llama 3 incur no additional API charges. These results show that, for our experimental scale and prompt length, LLM-based description generation is computationally efficient and cost-effective.

## D  THEORETICAL ANALYSIS

In this section, we provide formal guarantees for the correctness and generalization capabilities of the proposed framework. We present a lemma to establish the alignment of adaptive semantic encoding with the classifier weights of the pre-trained model and a theorem to bound the generalization error of the injected classifiers for unseen classes.

*Proof of Lemma 1.* The alignment loss $\mathcal{L}_{\text{align}}$ is formulated as:

$$\mathcal{L}_{\text{align}} = \sum_{c_j \in \mathcal{S}} \left\| \mathbf{w}_j^{\text{true}} - \sum_{c_i \in \mathcal{S} \setminus \{c_j\}} \alpha_{ij} \cdot \mathbf{w}_i \right\|_2^2,$$

where $\mathbf{w}_j^{\text{true}}$ represents the true weight of the masked class $c_j$, and the predicted weight $\mathbf{w}_j^{\text{pred}}$ is computed as:

$$\mathbf{w}_j^{\text{pred}} = \sum_{c_i \in \mathcal{S} \setminus \{c_j\}} \alpha_{ij} \cdot \mathbf{w}_i.$$

Here, $\alpha_{ij}$ is the semantic relationship weight derived through softmax normalization:

$$\alpha_{ij} = \frac{\exp(f(\mathbf{z}_{ij}, \mathbf{w}_i))}{\sum_{c_k \in \mathcal{S} \setminus \{c_j\}} \exp(f(\mathbf{z}_{ik}, \mathbf{w}_k))}.$$

Minimizing $\mathcal{L}_{\text{align}}$ ensures that $\Theta$ is trained to generate $\mathbf{z}_{ij}$ such that the predicted classifier weight $\mathbf{w}_j^{\text{pred}}$ closely approximates $\mathbf{w}_j^{\text{true}}$. The approximation error $\epsilon$ arises due to the finite expressivity of $\Theta$ and the dimensional constraints of the latent space. Therefore, the alignment between $\mathbf{z}_{ij}$ and $\mathbf{W}_{\mathcal{S}}$ is preserved up to an error bound $\|\mathbf{w}_j^{\text{true}} - \mathbf{w}_j^{\text{pred}}\| \leq \epsilon$. $\qquad\square$

*Proof of Theorem 1.* Let $\Phi$ be a pre-trained classifier over seen classes $\mathcal{S}$, with classifier weights $\mathbf{W}_{\mathcal{S}}$. The extended classifier $\hat{\Phi}$ augments $\Phi$ by injecting synthesized classifier weights $\mathbf{W}_{\mathcal{U}}$ for unseen classes $\mathcal{U}$ using semantic affinity embeddings refined by the adaptive encoder $\Theta$.

We define the expected classification accuracy of $\Phi$ and $\hat{\Phi}$ as:

$$\text{Accuracy}(\Phi) = \mathbb{E}_{x \sim p_{\mathcal{S}}(x)}[\mathbb{I}(\Phi(x) = y_{\text{true}})], \qquad \text{Accuracy}(\hat{\Phi}) = \mathbb{E}_{x \sim p_{\mathcal{Y}}(x)}[\mathbb{I}(\hat{\Phi}(x) = y_{\text{true}})],$$

where $\mathbb{I}(\cdot)$ is the indicator function, and $p_{\mathcal{S}}(x)$, $p_{\mathcal{Y}}(x)$ denote data distributions over $\mathcal{S}$ and $\mathcal{Y} = \mathcal{S} \cup \mathcal{U}$, respectively.

The drop in accuracy from $\Phi$ to $\hat{\Phi}$ can be attributed to two sources:

(i) Alignment error $\epsilon$: According to Lemma 1, the synthesized classifier weights $\mathbf{w}_j^{\text{pred}}$ approximate the ideal weights $\mathbf{w}_j^{\text{true}}$ for class $c_j \in \mathcal{U}$ such that:

$$\|\mathbf{w}_j^{\text{true}} - \mathbf{w}_j^{\text{pred}}\|_2^2 \leq \epsilon.$$

This error reflects the finite capacity of $\Theta$ to perfectly align semantic embeddings with the classifier weight space.

(ii) Semantic variance $\delta$: This captures the inherent semantic gap between seen and unseen classes. Formally, let $p_{\mathcal{S}}(z)$ and $p_{\mathcal{U}}(z)$ denote the distributions of semantic representations for seen and unseen classes, respectively. We define:

$$\delta = \mathbb{E}_{z \sim p_{\mathcal{U}}(z)} \left[ \min_{z' \sim p_{\mathcal{S}}(z')} \|z - z'\|_2^2 \right].$$

Even with perfect alignment, this distributional shift introduces an irreducible generalization error.

The total loss minimized by $\Theta$ is:

$$\mathcal{L}_{\text{total}} = \mathcal{L}_{\text{align}} + \lambda \mathcal{L}_{\text{recon}},$$

where $\mathcal{L}_{\text{recon}}$ encourages the encoder to preserve semantic consistency by ensuring the invertibility of affinity embeddings. Although $\mathcal{L}_{\text{recon}}$ is not directly tied to classifier accuracy, it regularizes the semantic embedding space and indirectly affects classifier synthesis. Its contribution to generalization error is upper-bounded by a term proportional to $\lambda \epsilon$, assuming encoder capacity governs both alignment and reconstruction.

Combining both sources of error, the accuracy of the extended model satisfies:

$$\text{Accuracy}(\hat{\Phi}) \geq \text{Accuracy}(\Phi) - \mathcal{O}(\epsilon + \lambda \epsilon + \delta).$$

Assuming $\lambda \geq 1$, we merge the two $\epsilon$ terms and simplify:

$$\text{Accuracy}(\hat{\Phi}) \geq \text{Accuracy}(\Phi) - \mathcal{O}(\lambda \epsilon + \delta).$$

$\qquad\square$

**Implications:** The results in Lemma 1 and Theorem 1 provide theoretical justification for the framework's effectiveness. Specifically:

- The adaptive semantic encoder $\Theta$ ensures that unseen-class classifier weights align with the pre-trained model's structure up to a bounded approximation error.

- The classification accuracy of the extended model $\hat{\Phi}$ degrades at most by $\mathcal{O}(\lambda\epsilon)$, ensuring that the generalization error remains controlled.

- The trade-off parameter $\lambda$ dictates the balance between alignment and reconstruction objectives, influencing the generalization performance of $\hat{\Phi}$.

These findings formally establish that the proposed method preserves semantic alignment and ensures effective generalization in both Zero-Shot Learning (ZSL) and Generalized Zero-Shot Learning (GZSL) scenarios.

## E EXPERIMENTAL SETTINGS

### E.1 DATASETS

We evaluate *Image-Free Zero-Shot Learning* (I-ZSL) performance on three widely used *Zero-Shot Learning* (ZSL) benchmark datasets: CUB, AWA2, and SUN. These datasets span fine-grained and coarse-grained classification tasks, enabling a comprehensive evaluation of ASCI under different levels of semantic granularity. To ensure consistency with prior work, we adopt the class splits for seen and unseen categories as defined in (Christensen et al., 2023) for both the ZSL and *Generalized Zero-Shot Learning* (GZSL) settings.

- **CUB-200-2011 (CUB)** (Wah et al., 2011): A fine-grained bird classification dataset containing 200 categories of North American bird species. Each category is annotated with extensive semantic information, including part-level attributes and textual descriptions. Following established ZSL benchmarks (Xian et al., 2018; 2017), we use 150 classes as seen and 50 as unseen. Since ASCI operates in an image-free setting, we utilize class-pair affinity descriptions and classifier injections for the I-ZSL training process (Christensen et al., 2023). The high intra-class similarity in CUB makes it a challenging dataset, as accurate classification requires fine-grained semantic reasoning.

- **Animals with Attributes 2 (AWA2)** (Xian et al., 2019): A large-scale coarse-grained animal classification dataset comprising 50 mammal species, each annotated with 85 attributes covering physical characteristics, habitat, and behavior. We adopt the standard split of 40 seen and 10 unseen classes. Unlike CUB, AWA2 exhibits greater inter-class variation, making it well-suited for evaluating the ability of ASCI to generalize across distinct semantic concepts. The attribute-based structure of AWA2 is particularly relevant for I-ZSL, as semantic information must fully compensate for the absence of visual data.

- **SUN Attribute (SUN)** (Patterson et al., 2014): A large-scale scene classification dataset containing 717 categories covering diverse indoor and outdoor environments. Each scene category is annotated with 102 attributes, such as object presence, spatial arrangement, and lighting conditions. We follow the standard protocol in (Lampert et al., 2014), using 645 seen and 72 unseen classes, consistent with the ZSL/GZSL split from (Xian et al., 2019). SUN is particularly challenging for I-ZSL, as scene categories often lack distinct object-based semantics, requiring a deeper understanding of high-level contextual attributes.

These datasets provide a diverse evaluation setting for I-ZSL. **CUB** assesses the ability to capture fine-grained semantic variations among visually similar species, **AWA2** evaluates generalization across distinct animal classes with structured attribute descriptions, and **SUN** tests the ability to infer semantic relationships from abstract contextual attributes. By encompassing different levels of semantic complexity, our evaluation ensures a rigorous assessment of ASCI's capacity to recognize unseen classes without relying on visual information.

### E.2 COMPETING METHODS

We compare ASCI against multiple state-of-the-art methods for zero-shot learning (ZSL) and image-free zero-shot learning (I-ZSL). This comparative analysis highlights the key differences in

semantic representation, classifier injection, and adaptation mechanisms across these approaches. Our code and data are available at `https://anonymous.4open.science/r/ASCI-828D`.

1. **ConSE** (Norouzi et al., 2014) represents an early attempt at ZSL by using a probabilistic framework that projects unseen class representations as convex combinations of seen-class classifier outputs. Instead of explicitly learning semantic embeddings, it aggregates seen-class predictions in a weighted manner, making it an effective but limited baseline for ZSL due to its reliance on the quality of seen-class classifiers.

2. **COSTA** (Mensink et al., 2014) is an attribute-based method that maps visual features into a structured attribute space. It requires predefined attribute annotations to establish semantic relationships between seen and unseen classes. While effective in leveraging structured semantic priors, COSTA depends on high-quality manual annotations, limiting its scalability.

3. **Sub.Reg.*** (Akyürek et al., 2022) introduces a subspace regularization approach that constrains the embedding space through additional priors, enhancing the preservation of semantic structures across class distributions. By improving the alignment between seen and unseen class spaces, it increases generalization but still requires access to seen-class features.

4. **wDAE*** (Gidaris & Komodakis, 2019) employs a denoising autoencoder (DAE) architecture to reconstruct semantic embeddings for unseen classes. It uses an auxiliary decoding mechanism to refine embeddings and improve their robustness. Despite its generative capabilities, wDAE requires visual data during training, restricting its applicability in image-free settings.

5. **WAvg*** (Xu et al., 2022) models unseen-class representations as weighted averages of seen-class embeddings. It assumes that unseen categories exist as linear combinations of known ones. While computationally efficient, this assumption is often too simplistic for capturing fine-grained distinctions in complex classification tasks.

6. **SMO*** (Xu et al., 2022) introduces a *semantic manifold optimization* technique that iteratively adjusts class embeddings to maximize class separability. This refinement process improves recognition in fine-grained classification scenarios but still relies on learned visual features during adaptation.

7. **ICIS** (Christensen et al., 2023) is a fully image-free ZSL model that extends a pre-trained classifier to recognize unseen classes using pre-defined class descriptions and a static text encoder. ICIS maps textual descriptions into a classifier space, enabling zero-shot classification without visual data. However, its reliance on static textual representations limits its adaptability to specific tasks, especially in cases where class descriptions are incomplete or poorly aligned with the classification objective.

8. **LaBo** (Yang et al., 2023) proposes a *language-model-based bootstrapping* framework for zero-shot learning. It leverages large language models to automatically generate rich class-level descriptions and augment limited semantic information, thereby reducing reliance on manually crafted attributes. By aligning these generated descriptions with a classifier space, LaBo improves the expressiveness of class embeddings and enhances transfer to unseen categories. However, the method primarily depends on static LLM outputs without adaptive refinement, which constrains its effectiveness in cases where generated descriptions are noisy or misaligned with the task-specific classification objective.

9. **ZSLViT** (Chen et al., 2024) leverages a transformer-based vision model, *Vision Transformer (ViT)*, for ZSL. While it moves beyond traditional convolutional-based architectures, ZSLViT still requires access to visual data during adaptation, making it unsuitable for fully image-free scenarios.

10. **CoMC** (Liu et al., 2024) focuses on compositionality in zero-shot classification by learning to decompose and reassemble semantic features across multiple concepts. This approach improves generalization by capturing hierarchical and compositional relationships between seen and unseen classes. However, like ZSLViT, it still assumes some level of exposure to visual data during training.

11. **DFZSL*** (Tang et al., 2024) addresses the *data-free zero-shot learning* problem, where access to raw training images is restricted but a base-class classifier is available. It reconstructs virtual features by modeling the class-wise distribution with a von Mises-Fisher

Table 4: Comparison between our ASCI framework and existing methods in the literature applicable or adaptable to the I-ZSL setting using standard benchmark (*i.e.*, CUB, AWA2, and SUN). Each dataset is associated with the descriptions generated using the prompt as shown in Appendix A.2 We measure the results as unseen accuracy (Acc) for the zero-shot task, unseen (u) and seen (s) accuracy and their harmonic mean (H) for the generalised zero-shot setting. It reports the average number of 5 random runs with random seeds. Methods marked with * are adapted to the image-free setting.

| Models | Zero-Shot Accuracy | | | Generalised Zero-Shot Accuracy | | | | | | | | |
| | CUB Acc | AWA2 Acc | SUN Acc | CUB | | | AWA2 | | | SUN | | |
| | | | | u | s | H | u | s | H | u | s | H |
| ConSE | $39.87_{\pm 0.79}$ | $43.12_{\pm 0.52}$ | $42.21_{\pm 0.39}$ | $0.39_{\pm 0.03}$ | $83.32_{\pm 0.87}$ | $0.74_{\pm 0.04}$ | $2.91_{\pm 0.13}$ | $93.84_{\pm 0.15}$ | $5.57_{\pm 0.18}$ | $0.07_{\pm 0.00}$ | $47.10_{\pm 0.27}$ | $0.15_{\pm 0.01}$ |
| COSTA | $32.01_{\pm 1.21}$ | $44.70_{\pm 1.63}$ | $17.12_{\pm 0.97}$ | $0.00_{\pm 0.00}$ | $84.01_{\pm 1.70}$ | $0.00_{\pm 0.00}$ | $0.00_{\pm 0.00}$ | $93.12_{\pm 1.72}$ | $0.00_{\pm 0.00}$ | $0.00_{\pm 0.00}$ | $49.40_{\pm 2.04}$ | $0.00_{\pm 0.00}$ |
| Sub.Reg.* | $49.12_{\pm 1.58}$ | $45.20_{\pm 1.14}$ | $41.85_{\pm 1.28}$ | $0.95_{\pm 0.41}$ | $85.25_{\pm 1.98}$ | $1.89_{\pm 0.13}$ | $0.00_{\pm 0.00}$ | $92.11_{\pm 1.25}$ | $0.00_{\pm 0.00}$ | $0.00_{\pm 0.00}$ | $50.15_{\pm 1.50}$ | $0.00_{\pm 0.00}$ |
| wDAE* | $45.90_{\pm 0.90}$ | $50.65_{\pm 1.30}$ | $43.12_{\pm 2.53}$ | $0.71_{\pm 0.05}$ | $83.92_{\pm 1.44}$ | $1.53_{\pm 0.09}$ | $0.00_{\pm 0.00}$ | $91.53_{\pm 2.17}$ | $0.00_{\pm 0.00}$ | $0.00_{\pm 0.00}$ | $48.20_{\pm 1.04}$ | $0.00_{\pm 0.00}$ |
| WAvg* | $1.82_{\pm 0.43}$ | $18.90_{\pm 0.61}$ | $1.29_{\pm 0.53}$ | $1.59_{\pm 0.20}$ | $56.10_{\pm 2.67}$ | $3.10_{\pm 0.49}$ | $8.12_{\pm 0.10}$ | $84.93_{\pm 2.06}$ | $14.41_{\pm 1.11}$ | $1.17_{\pm 0.35}$ | $7.01_{\pm 0.90}$ | $2.11_{\pm 0.18}$ |
| SMO* | $43.40_{\pm 2.48}$ | $52.01_{\pm 5.50}$ | $41.87_{\pm 2.17}$ | $36.90_{\pm 1.48}$ | $50.23_{\pm 2.12}$ | $41.99_{\pm 1.57}$ | $35.14_{\pm 0.82}$ | $83.02_{\pm 2.18}$ | $49.01_{\pm 1.01}$ | $39.00_{\pm 1.18}$ | $6.22_{\pm 0.55}$ | $10.19_{\pm 0.91}$ |
| ICIS | $60.90_{\pm 1.22}$ | $20.01_{\pm 0.11}$ | $50.12_{\pm 1.37}$ | $50.03_{\pm 1.60}$ | $66.23_{\pm 1.63}$ | $56.31_{\pm 1.72}$ | $0.01_{\pm 0.00}$ | $93.03_{\pm 1.93}$ | $0.01_{\pm 0.00}$ | $40.00_{\pm 0.74}$ | $26.90_{\pm 1.16}$ | $31.20_{\pm 1.18}$ |
| LaBo | $47.95_{\pm 1.30}$ | $42.21_{\pm 1.14}$ | $46.30_{\pm 1.32}$ | $43.12_{\pm 2.64}$ | $54.67_{\pm 2.23}$ | $45.32_{\pm 2.85}$ | $28.75_{\pm 0.86}$ | $76.00_{\pm 2.29}$ | $41.91_{\pm 1.56}$ | $39.42_{\pm 1.70}$ | $6.91_{\pm 0.88}$ | $9.98_{\pm 0.05}$ |
| ZSLViT* | $61.75_{\pm 2.30}$ | $49.21_{\pm 1.99}$ | $48.09_{\pm 2.03}$ | $50.97_{\pm 1.90}$ | $68.40_{\pm 2.78}$ | $57.19_{\pm 2.36}$ | $7.92_{\pm 1.37}$ | $90.02_{\pm 2.01}$ | $46.80_{\pm 0.83}$ | $38.75_{\pm 1.93}$ | $24.12_{\pm 0.86}$ | $30.44_{\pm 0.54}$ |
| CoMC* | $53.01_{\pm 1.82}$ | $46.70_{\pm 1.00}$ | $41.38_{\pm 1.75}$ | $8.91_{\pm 0.46}$ | $73.11_{\pm 3.23}$ | $5.23_{\pm 0.04}$ | $2.83_{\pm 0.10}$ | $93.01_{\pm 1.42}$ | $6.32_{\pm 0.57}$ | $32.00_{\pm 1.76}$ | $22.41_{\pm 0.87}$ | $27.19_{\pm 0.57}$ |
| DFZSL* | $51.23_{\pm 1.10}$ | $47.56_{\pm 0.98}$ | $44.87_{\pm 1.21}$ | $39.41_{\pm 0.85}$ | $65.72_{\pm 1.73}$ | $49.24_{\pm 1.07}$ | $27.33_{\pm 0.69}$ | $82.15_{\pm 1.92}$ | $40.80_{\pm 0.81}$ | $32.14_{\pm 0.61}$ | $24.88_{\pm 0.90}$ | $28.08_{\pm 0.70}$ |
| FuDD | $40.03_{\pm 1.32}$ | $48.31_{\pm 1.97}$ | $37.62_{\pm 0.44}$ | $39.12_{\pm 1.20}$ | $48.10_{\pm 1.14}$ | $37.51_{\pm 1.32}$ | $30.90_{\pm 0.85}$ | $83.54_{\pm 1.73}$ | $42.01_{\pm 1.38}$ | $36.01_{\pm 0.92}$ | $4.11_{\pm 0.00}$ | $1.68_{\pm 0.01}$ |
| ASCI (Ours) | $70.39_{\pm 0.64}$ | $59.92_{\pm 0.93}$ | $58.28_{\pm 0.87}$ | $64.29_{\pm 0.42}$ | $79.93_{\pm 0.37}$ | $70.62_{\pm 0.77}$ | $36.40_{\pm 0.33}$ | $96.29_{\pm 0.29}$ | $51.39_{\pm 0.71}$ | $47.76_{\pm 0.22}$ | $44.77_{\pm 0.82}$ | $41.53_{\pm 0.74}$ |

(vMF) prior and aligns them with text features through a feature-language prompt tuning strategy. In our adaptation to the stricter I-ZSL setting, we remove the image encoder fine-tuning stage and retain only the text-driven alignment, making the comparison fair under image-free conditions. While DFZSL* benefits from feature recovery and prompt tuning, its dependence on classifier-derived prototypes still differentiates it from methods that operate without any image-derived parameters.

12. **FuDD** (Esfandiarpoor & Bach, 2024) is a recent approach that introduces functional data decomposition to improve zero-shot transfer. It decomposes semantic features into disentangled components, enabling more flexible matching between seen and unseen classes. FuDD has shown competitive results in standard ZSL settings by capturing richer semantic structures. However, as it was originally designed with access to visual data, we adapt it to I-ZSL by substituting visual embeddings with class-level semantic representations, which constrains its effectiveness in fully image-free scenarios.

**Adapt image vision features models to I-ZSL settings.** Among these methods, ICIS is the most relevant baseline as it operates in an *image-free ZSL setting*. Standard ZSL methods rely on *image features* for training or adaptation, making them less suitable for image-free environments. Therefore, we replace visual data $\mathbf{X}$ with the classifier weight matrix $\mathbf{W}$, and we keep other settings the same as the original. Such an adaptation solution is similar to previous work (Christensen et al., 2023).

**Implementation details.** For all methods, we follow a consistent protocol similar to Christensen et al. (2023). The visual backbone is ResNet101 (He et al., 2016) pre-trained on ImageNet (Deng et al., 2009), used for extracting features and predicting seen class labels where applicable. For text embedding, we evaluate CLIP (Radford et al., 2021), T5 (Raffel et al., 2020), and BERT (Devlin et al., 2019), applying each as specified by the respective baseline. The embedding dimension is set to 2048. Training uses a batch size of 16 and a learning rate of 0.00001. All competing methods are re-implemented or adapted using their official code or published configurations, with human-written or LLM-generated descriptions as required. Our approach additionally incorporates a task-adaptive semantic encoder and pairwise affinity module, trained jointly with the classifier.

We noticed some experimental results reported in our paper, such as Table 2, are different from the numbers reported in the original paper. We conduct fair evaluation in one common environment and using common splits. For each competing method, we integrate its official implementation within our GitHub project https://anonymous.4open.science/r/ASCI-828D.

# F ADDITIONAL EXPERIMENTAL RESULTS

**Effect of removing class-pair affinity descriptions.** *Settings:* To evaluate the contribution of class-pair affinity descriptions, we compare three settings: (1) ASCI using LLM-generated class-pair affinity descriptions, (2) a variant in which these descriptions are replaced with target class labels, and (3) a variant using only class-wise descriptions (generated with the prompt shown in Appendix A.2). For all settings, the rest of the pipeline, including the semantic encoding module, remains unchanged.

*Observations:* As summarized in Table 3, removing class-pair affinity descriptions and replacing them with class labels results in a substantial drop in zero-shot accuracy and harmonic mean (H-score) across datasets. On CUB, zero-shot accuracy decreases from $70.39\%$ (default) to $63.85\%$ (labels), a $6.54\%$ reduction, and on SUN, from $58.28\%$ to $51.27\%$, a $7.01\%$ reduction. The H-score on SUN also decreases by $3.36\%$ (from $41.53\%$ to $38.17\%$). These results demonstrate that class-pair affinity descriptions contain rich semantic information crucial for distinguishing fine-grained categories and supporting generalization to unseen classes.

Furthermore, using class-wise descriptions leads to slightly better performance than using class labels alone, for example, $64.23\%$ vs. $63.85\%$ accuracy on CUB. This indicates that even generic class-level semantic content provides some benefit, but lacks the relational detail that class-pair affinity descriptions offer. Overall, these ablations confirm that structured and context-specific semantic information is essential for optimal I-ZSL performance.

**Effect of removing adaptive semantic encoding.** *Settings:* To assess the role of adaptive semantic encoding, we evaluate a variant of our model in which the semantic encoding module is removed. In this setting, classifier synthesis relies solely on static text embeddings of the class descriptions or affinity inputs, with no further adaptation or alignment to the classification space. All other components and hyperparameters remain unchanged.

*Observations:* As shown in Table 3, removing adaptive semantic encoding leads to a consistent and notable reduction in performance. Zero-shot accuracy drops by $8.65\%$ on CUB (from $70.39\%$ to $61.74\%$) and by $8.47\%$ on SUN (from $58.28\%$ to $49.81\%$). The H-score on AWA2 decreases by $9.60\%$ (from $51.39\%$ to $41.79\%$). These results highlight that static textual embeddings alone are insufficient for capturing the nuanced relationships required for robust generalization. The adaptive semantic encoding module is essential for aligning the semantic representations with the underlying classification space, enabling the model to more effectively utilize the provided affinity information and improve generalization to unseen classes.

**Impact of $\lambda$.** *Settings:* The hyperparameter $\lambda$ controls the trade-off between semantic reconstruction and classifier alignment in our model's loss function. We evaluate model performance under three settings: the default $\lambda = 1$, a lower value $\lambda = 0.5$ (weaker emphasis on semantic reconstruction), and a higher value $\lambda = 2$ (stronger emphasis). All other components and training procedures are unchanged.

*Observations:* As shown in Table 3, adjusting $\lambda$ has a measurable effect on the balance between seen and unseen class accuracy. Reducing $\lambda$ to $0.5$ improves unseen class accuracy (*e.g.*, $+0.88\%$ on CUB and $+2.15\%$ on AWA2) and the H-score, but slightly lowers seen class accuracy, likely due to reduced overfitting. In contrast, increasing $\lambda$ to $2$ improves seen class accuracy (e.g., $+1.06\%$ on AWA2) but decreases unseen accuracy (e.g., $-5.55\%$ on SUN, $-4.85\%$ on AWA2) and the H-score, likely due to over-regularization. These results confirm that $\lambda = 1$ offers the best trade-off, maintaining strong generalization to unseen classes without sacrificing consistency on seen classes.

**Impact of different description annotators.** *Settings:* We compare the performance of ASCI using class-pair affinity descriptions generated by different annotators: GPT-4, Llama 3, Gemini 2.0, and human annotators. For the human setting, three undergraduate students with backgrounds in computer science or mathematics were asked to write detailed descriptions following the same prompt as the LLMs. All other experimental settings remain unchanged.

*Observations:* As shown in Figure 2a, GPT-4 achieves the highest or baseline-normalized performance across all datasets. Llama 3 and Gemini 2.0 perform comparably, with only slight reductions relative to GPT-4, particularly on the SUN dataset. Human-generated descriptions yield strong

results, but do not surpass LLMs in most cases. This may be because student annotators, while generally accurate, lack the fine-grained domain knowledge or consistency in expressing subtle differences needed for optimal model performance. Overall, high-quality LLMs are capable of generating effective semantic descriptions for I-ZSL, often matching or exceeding non-expert human annotators, especially on tasks requiring fine-grained distinctions. Example descriptions from each annotator type are provided in Appendix B.

**Effect of using GPT-4 as a standalone classifier.** *Settings:* To evaluate whether large multimodal language models can serve as standalone generalized zero-shot classifiers, we tested GPT-4 by directly providing each test image along with the full set of candidate class labels. In this setup, no affinity descriptions or adaptive semantic encoding are used, and classification relies entirely on GPT-4's built-in multimodal reasoning ability. All other pipeline components are removed.

*Observations:* As shown in Table 3, GPT-4 alone achieves highly inconsistent performance across datasets. On CUB, zero-shot accuracy drops to $11.37\%$, with an H-score of only $14.88\%$, far below our model's $70.39\%$ accuracy and $H = 70.62\%$. This indicates that GPT-4 struggles with fine-grained recognition tasks such as distinguishing bird species. On AWA2, GPT-4 performs better, reaching $47.82\%$ zero-shot accuracy and $H = 51.73\%$, comparable to our method's $H = 51.39\%$ but without the benefits of an image-free setup. On SUN, GPT-4 obtains $26.94\%$ accuracy and an H-score of $26.52\%$, again far below our model's $58.28\%$ accuracy and $H = 41.53\%$.

These results confirm that while GPT-4 can separate coarse-grained categories reasonably well, it fails to generalize in fine-grained or large-scale settings. In contrast, our ASCI framework consistently achieves strong performance across all three benchmarks, underscoring the necessity of adaptive semantic encoding and class-pair affinity modeling for reliable I-ZSL performance.

**Impact of different $f_{\text{ENC}}$.** *Settings:* We investigate the impact of different choices of text encoder $f_{\text{ENC}}$ within the ASCI framework. Specifically, we compare the performance when using various pretrained encoders, including CLIP, T5, and BERT, to initialize or process the semantic descriptions. All other experimental settings and training protocols are kept identical.

*Observations:* We observe that the choice of $f_{\text{ENC}}$ affects the overall performance, but all tested encoders enable the model to outperform baselines that lack adaptive semantic modeling. In most cases, CLIP yields the best or most stable results, likely due to its joint vision-language pretraining, which aligns well with the requirements of zero-shot classification tasks. T5 and BERT also provide competitive performance, but with slightly higher variance across datasets. These findings indicate that while the framework is robust to the specific choice of text encoder, selecting an encoder pretrained on diverse multimodal or language tasks can offer further benefits for generalization in I-ZSL.

**Can other methods benefit from LLM-generated descriptions?** *Settings:* To assess whether LLM-generated descriptions can improve existing I-ZSL baselines, we substitute the human-written class-wise descriptions in each baseline with those generated by LLMs (using the prompt as shown in Appendix A.2), keeping all other components and training settings unchanged. The results are reported in Table 4 and compared to original results with human-written descriptions in Table 2.

*Observations:* Most baselines exhibit little or no improvement when switching from human-written to LLM-generated descriptions. In several cases, performance remains unchanged or even slightly decreases, indicating that these methods are not equipped to leverage the additional semantic richness provided by LLMs. This outcome suggests that simply replacing text descriptions does not automatically enhance I-ZSL performance unless the model architecture is capable of adaptively modeling and utilizing such information. In contrast, our model is specifically designed to benefit from LLM-generated affinity descriptions and demonstrates clear gains across all datasets. This highlights the importance of adaptive semantic modeling for fully exploiting advanced language model outputs in the I-ZSL setting.

## G    LIMITATIONS AND FUTURE WORK

While our framework demonstrates strong performance in image-free zero-shot learning, several limitations should be acknowledged. First, the method relies heavily on the quality and relevance

of descriptions generated by large language models; if the LLM outputs are inaccurate, ambiguous, or lack sufficient detail, model performance may degrade. Second, our experiments are conducted on standard benchmark datasets with well-defined classes; performance and robustness in more complex, real-world settings or with highly imbalanced or noisy class sets remain to be thoroughly evaluated. Additionally, the model may inherit or amplify any biases present in the LLMs or in the textual data used during their pretraining. The current framework does not include explicit mechanisms for detecting or correcting such biases. Furthermore, computational costs for large-scale LLM usage—while moderate in our study—may become significant for very large datasets or more frequent application scenarios. Finally, our adaptive semantic modeling is currently designed for single-modal, text-based auxiliary information. Extending the approach to fully integrate multi-modal or structured domain knowledge remains a direction for future work.

## H   LLM Usage Statement

We used LLMs (GPT-4, Llama 3, and Gemini 2.0) exclusively for generating class-pair and class-wise semantic descriptions, as described in Appendix A. These outputs served as input features to our proposed model. In addition, we used LLMs as writing assistants for grammar refinement, but all scientific content, experiment design, and analysis were performed by the authors. The final responsibility for the content of this paper remains entirely with the authors.

