# OpenReview forum: "Image-Free Zero-Shot Learning via Adaptive Semantic-Guided Classifier Injection"
_ICLR.cc/2026/Conference — ICLR 2026 Conference Withdrawn Submission_

### Official Review · Reviewer_FNmo · 2025-10-28

**Soundness:** 2
**Presentation:** 3
**Contribution:** 3
**Rating:** 4
**Confidence:** 5

**Summary:**

This article proposes an adaptive semantic guided classifier injection (ASCI) method to recognize invisible classes in an image free manner (zero shot learning, ZSL). Specifically, the proposed method firstly uses the large language models (LLMs) to generate  class-pair affinity descriptions, which are refined by a trainable text encoder. Next, dynamically computed affinity scores is leveraged to keep the structural consistency of the pre-trained classification space.

**Strengths:**

Strengths:

S1: The method used in this paper is quite innovative, especially in generating 'category-paired descriptions'. I think compared to generating individual category descriptions, generating paired category descriptions can indeed amplify the differences between categories.

S2: This paper is well expressed. The structure is very clear, and the method types have been clearly sorted out.

S3: Compared to ICIS (the most important baseline), the performance has significantly improvement.

**Weaknesses:**

Major Weaknesses:

**W1: Overclaimed contributions.**

The method proposed in this paper is indeed innovative, but it does not make as significant a contribution as claimed.
Firstly, in Line 060 to 062, this paper claims that

_"**Unlike previous work, ASCI eliminates the dependency on pre-defined
descriptions** and instead leverages large language models to generate class-pair affinity descriptions that capture structured relationships between seen and unseen classes. "_

However, this matter is one of the main breakthrough points in the current ZSL work.
For example, I2DFormer [1] utilizes a bimodal transformer to adaptively generate category semantic encoding.
VGSE [2] utilizes visual self supervision to generate visual encoding.‘
More recently, I2MVFormer [3] also uses LLM to generate category text, and InfZSL also uses LLM to generate category concepts.
But this work did not mention these works at all and discuss comparisons.

Secondly, in Line 063 to 065, this paper claims that a trainable text encoder is important in ZSL. However, basically, all transformer-based zsl methods [1] [3] [4] utilize trainable text encoders to generate class semantics.

**In short, the two improvements proposed in this work (getting rid of human descriptions and  trainable text encoder) are indeed innovative, but previous zsl research has a long history in both areas.**
I think this job requires further in-depth research and comparative experiments on these two points to explore more insights and make solid contributions

**W2: Definition issue for Image-free ZSL (I-ZSL).**

In table 1, the authors classify existing methods accoding to three factors: no visual data, no human descriptions, and adaptive semantic guidance. This is quite far fetched. Because in previous classic works ([5] and ICIS), only the first point must be met, and the latter two seem to be forcibly classified to highlight the method proposed in this article.
More importantly, it includes some methods for non-ZSL methods, such as LabO that is not a ZSL method and does not meet the ZSL definition in the section 3.1 of this paper.

**W3: Missing some very close SOTAs that I have mentioned in W1 and W2.**

**W4:** As is well known, LLMs have 'hallucination' problems and InfZSL [6] indicates that LLM often generates non-visual category descriptions. Has this work addressed this issue?

Minor Weaknesses:

**W5:** Some abbreviations lack the full name, e.g., "AI" in Line 30.


[1] Naeem M F, Xian Y, Gool L V, et al. I2dformer: Learning image to document attention for zero-shot image classification[J]. Advances in Neural Information Processing Systems, 2022, 35: 12283-12294.

[2] Xu W, Xian Y, Wang J, et al. Vgse: Visually-grounded semantic embeddings for zero-shot learning[C]//Proceedings of the IEEE/CVF conference on computer vision and pattern recognition. 2022: 9316-9325.

[3] Naeem M F, Khan M G Z A, Xian Y, et al. I2mvformer: Large language model generated multi-view document supervision for zero-shot image classification[C]//Proceedings of the IEEE/CVF Conference on Computer Vision and Pattern Recognition. 2023: 15169-15179.

[4] Qu X, Yu J, Gai K, et al. Visual-semantic decomposition and partial alignment for document-based zero-shot learning[C]//Proceedings of the 32nd ACM International Conference on Multimedia. 2024: 4581-4590.

[5] Selvaraju R R, Chattopadhyay P, Elhoseiny M, et al. Choose your neuron: Incorporating domain knowledge through neuron-importance[C]//Proceedings of the European conference on computer vision (ECCV). 2018: 526-541.

[6] Ye Z, Gowda S N, Chen S, et al. Interpretable Zero-shot Learning with Infinite Class Concepts[J]. arXiv preprint arXiv:2505.03361, 2025.

**Questions:**

Q1:

In abs., the authors claim the method can achieve "robust classifiers", but I cannot find related definitions or experiments to tie the claim. Furthermore, what is relative to 'robust'? Adversaroal robustness? Imbalanced/long-tailed robustness? Data-scarcity robustness? or Data-corruption robustness?

Q2:

The LaBo is not a ZSL method, so how was the performance of this method measured in Table2?

Q3:

A classic problem in ZSL.
Because the trained text encoder is only trained on seen classes, will the obtained class semantics contain seen-unseen bias?

---

### Official Review · Reviewer_y2BH · 2025-10-30

**Soundness:** 3
**Presentation:** 3
**Contribution:** 3
**Rating:** 4
**Confidence:** 3

**Summary:**

The paper addresses the limitations of existing I-ZSL approaches, which rely on predefined class descriptions and static text encoders. It proposes an Adaptive Semantic-Guided Classifier Injection (ASCI) framework that leverages large language models to generate class descriptions, trains an adaptive text encoder, and dynamically injects classifiers. Experiments on benchmark datasets such as CUB, AWA2, and SUN demonstrate superior performance, particularly in fine-grained classification tasks. The study is well-motivated by real-world scenarios such as privacy-restricted settings, and it is logically coherent and well written.

**Strengths:**

1.The study directly targets the core limitations of existing I-ZSL methods that rely on manually annotated descriptions and static encoders, and proposes a corresponding solution.
2.The proposed approach fulfills the three key requirements of I-ZSL—independence from visual data, freedom from manually crafted class descriptions, and adaptive semantic guidance. By leveraging class descriptions to capture inter-class relationships and an adaptive encoder to align task semantics, it effectively overcomes the constraints of traditional methods.
3.The experiments cover both conventional and generalized ZSL settings, comparing the proposed method with 12 mainstream baselines. Extensive ablation studies validate the necessity of key components, and further analyses explore the effects of hyperparameters, text encoders, and description generators.

**Weaknesses:**

1.The method heavily relies on the accuracy and completeness of the descriptions produced by large language models. If the LLM outputs are biased, ambiguous, or contain missing information, the model’s performance may degrade significantly. The current version lacks an effective mechanism for correcting or optimizing the quality of these generated descriptions.
2.The framework does not incorporate any detection or mitigation strategies for biases that may exist in the LLM’s pretraining data. Consequently, the generated descriptions might inherit such biases, potentially affecting the fairness of the model’s classification outcomes.

**Questions:**

1.Does the use of fixed prompts for LLM-generated class-pair descriptions ensure semantic accuracy across different domains? To what extent does the variation in LLM generation quality affect overall model performance?
2.During the joint training of the adaptive semantic encoder, do the gradients of the two loss components interfere with each other? Is the masked class-weight computation strategy influenced by the number of seen classes?
3.How does the number of class pairs affect the model’s training efficiency and accuracy? Is there an optimal sampling strategy to balance performance and computational efficiency?

---

### Official Review · Reviewer_AD1Y · 2025-11-01

**Soundness:** 3
**Presentation:** 4
**Contribution:** 3
**Rating:** 6
**Confidence:** 4

**Summary:**

This paper studies the problem of image-free zero-shot learning, where no visual data are available for unseen classes. The authors introduce ASCI, a framework that uses large language models to describe relationships between class pairs and a small adaptive encoder to turn these relations into classifier weights for unseen categories. The method builds on prior work such as ICIS but removes the need for manual text descriptions and fixed encoders. Experiments on CUB, AWA2, and SUN show consistent improvements over previous baselines, and several ablation studies verify the main design choices.

**Strengths:**

1. The paper presents a creative approach that learns class-pair semantic affinities and maps them adaptively into the classifier space, which proves to be both novel and practically valuable.
2. The authors conduct comprehensive experiments on three benchmarks with twelve baselines and multiple ablation studies, demonstrating the robustness and effectiveness of the proposed method.
3. The paper is clearly structured, with well-organized figures and detailed explanations, which enhance transparency and ease of understanding even without direct code access.
4. The study maintains a clear link between problem motivation and implementation, resulting in a contribution that is both conceptually accessible and technically solid.

**Weaknesses:**

1. In the method description, the paper mentions that under the image-free setting the model can still access the classifier weights of seen categories while no visual features are available. This makes the approach more of a model-accessible but data-free setting rather than a fully data-free one. Clarifying this distinction would help readers understand the intended application scope more clearly.
2. All experiments are conducted on the common I-ZSL datasets (CUB, AWA2, SUN) following previous work. These results convincingly show that the method works well under standard conditions, though they leave open how it would behave on datasets with larger domain gaps or more complex label spaces. Expanding the evaluation in future work would make the claims more general.
3. The method constructs and encodes relations for each pair of classes and aggregates them into an affinity structure. While this design is reasonable for current benchmarks, the paper does not discuss how the computation would scale to larger label sets or whether approximations are needed. A brief discussion would improve completeness.

**Questions:**

1. ASCI assumes access to the classifier layer of a pre-trained model while images are unavailable. Could the authors briefly clarify what kind of practical situations this setting is meant to represent, and how it differs from typical zero-shot or data-free learning setups?
2. The method relies on LLM-generated class-pair relations to guide classifier construction. It would be interesting to understand whether these relational descriptions capture mainly hierarchical, visual, or functional similarities, and which of them contribute most to performance.
3. It would be helpful to know whether the performance of ASCI changes notably with different qualities or styles of LLM-generated affinity descriptions.

---

### Official Review · Reviewer_8Liy · 2025-11-01

**Soundness:** 2
**Presentation:** 2
**Contribution:** 2
**Rating:** 2
**Confidence:** 3

**Summary:**

This paper addresses image-free zero-shot learning (i-ZSL), where no images of unseen classes are available during training. The method builds on Christensen et al. (ICCV’23), which learns to regress linear classifier weights from textual class embeddings. This submission instead estimates the classifier for an unseen class as a linear combination of seen-class classifiers, where the coefficients are computed from LLM-derived class-pair affinity descriptions (new class to each seen classes). Training uses seen classes in a leave-one-out manner, with an autoencoder-style regularization to reconstruct affinity scores. Experiments are reported on CUB, SUN, and AWA2 in both ZSL and GZSL settings.

**Strengths:**

- Nice Intuitive formulation: unseen classifier as a mixture of seen-class classifiers guided by LLM-produced affinities.
- Uses LLM semantic knowledge in a principled way.
- Reported results are strong relative to prior i-ZSL baselines on standard benchmarks.

**Weaknesses:**

- Inference ambiguity: the paper appears to claim classification without using image features at test time. This requires clarification (see Questions).
- Theory section (Sec. 3.6): Lemma 1 is trivial from the definition of the loss; proofs are informal and skip necessary assumptions.
- Result presentation: tables (especially Table 2) are overcrowded and text is very small, making them difficult to read.
- Fairness of comparisons: replacing image features with classifier weights to shoehorn image-based methods does not seem like a fair and meaningful comparison.

Correctness
- Method seems plausible, but the inference-time description appears internally inconsistent: the logits as written do not depend on the input image.
- Theory has correctness issues: Lemma 1 is tautological, and the theorem depends on assumptions about data distributions that are not stated.

**Questions:**

A. Inference-time pipeline (critical)
The paper states (paraphrasing):

Since the feature extractor F is inaccessible, classification is performed in semantic space.
For a query instance, logits are computed as
$f_S = W_S^T Z,   f_U = W_U^T Z$,
where Z represents semantic embeddings derived from class-pair affinity descriptions.

1. If Z is derived from class-pair text descriptions, it is class-dependent but not image-dependent. How can logits differ across test images?
2. Is the intended inference rule actually
f_c(x) = w_c^T \phi(x)
where \phi(x) is an image feature extractor that is not trained further?
3. If no image features are used at test time, what object is being classified? Please clarify and correct the description.

B. Section 3.6 (theory)
1. Lemma 1: Given that $\mathcal{L}_{\mathrm{align}}$ is defined as a sum of non-negative terms, the statement is tautological. The proof in the appendix does not reference $\mathcal{L}_{\mathrm{align}}$ at all. Please restate and prove rigorously.

2. For the accuracy bound theorem:

- What are the assumptions on the data distribution (\phi(x), y)
- Where does the equation on line 964 come from?
- Is the left-hand side guaranteed to be non-negative?

---

### Note · Authors · 2025-12-03

**Comment:**

Thanks for the reviews. We have decided to withdraw this paper for revision.

**Withdrawal Confirmation:**

I have read and agree with the venue's withdrawal policy on behalf of myself and my co-authors.